# Gender Differences in the Risk for Incident Non-Alcoholic Fatty Liver Disease According to the Transition of Abdominal Obesity Status: A 16-Year Cohort Study

**DOI:** 10.3390/nu15132880

**Published:** 2023-06-25

**Authors:** Jun-Hyuk Lee, Soyoung Jeon, Hye Sun Lee, Yu-Jin Kwon

**Affiliations:** 1Department of Family Medicine, Nowon Eulji Medical Center, Eulji University School of Medicine, Seoul 01830, Republic of Korea; swpapa@eulji.ac.kr; 2Department of Medicine, Hanyang University Graduate School of Medicine, Seoul 04763, Republic of Korea; 3Biostatistics Collaboration Unit, Department of Research Affairs, Yonsei University College of Medicine, Seoul 03277, Republic of Korea; jsy0331@yuhs.ac; 4Department of Family Medicine, Yongin Severance Hospital, Yonsei University College of Medicine, Yongin 16995, Republic of Korea

**Keywords:** abdominal obesity, waist circumference, non-alcoholic fatty liver disease

## Abstract

Waist circumference (WC) is an important predictor of long-term adverse outcomes. We aimed at assessing the correlation between abdominal obesity (AO) patterns and non-alcoholic fatty liver disease (NAFLD). Data from 4467 adults aged 40–69 years and without NAFLD who participated in the Korean Genome and Epidemiology Study were analyzed. Participants were classified according to two-year WC pattern into four groups: persistent lean WC, improved AO, progressed to AO, and persistent AO. NAFLD was defined as NAFLD–liver fat score >−0.640. Multiple Cox proportional hazards regression analysis revealed that the fully adjusted hazard ratio (HR) (95% confidence intervals (CIs)) for NAFLD in persistent AO, progressed to AO, and improved AO groups compared to the persistent lean WC group was 1.33 (1.13–1.57), 1.73 (1.48–2.02), and 1.06 (0.84–1.33), respectively. Women in persistent AO or progressed to AO groups had significantly higher risk for NAFLD than those in persistent lean WC or improved AO groups. Men who had progressed to an AO event over two years had significantly higher risk for NAFLD than those without any AO event over two years. Maintaining lean WC and improving AO would be successful strategies for preventing NAFLD in women, while maintaining lean WC would be more effective in men.

## 1. Introduction

Non-alcoholic fatty liver disease (NAFLD) refers to the occurrence of hepatic steatosis without secondary causes such as excessive alcohol intake, viral hepatitis, autoimmune hepatitis, or medications. NAFLD not only induces liver-related complications, such as hepatic fibrosis, liver cirrhosis, and hepatocellular carcinoma, but also increases the risk for cardiovascular diseases (CVDs). In addition to its clinical importance, the global prevalence of NAFLD was estimated at 24% in 2015 and 37.3% in 2019, while the national prevalence in Korea is estimated to be between 29.0% and 31.0% over the last decade [1,2,3]. Therefore, the prevention and management of NAFLD has been emphasized. Strong associations exist among insulin resistance, the presence of visceral fat, and NAFLD. Excessive visceral adipose tissue (VAT) promotes secretion of pro-inflammatory cytokines such as interleukin 6 (IL-6) and tumor necrosis factor α (TNF-α), which leads to the development and worsening of insulin resistance [4]. Insulin resistance upregulates the expression of hormone-sensitive lipase in adipocyte, especially visceral fat, which hydrolyzes triglyceride into glycerol and fatty acids [5]. Free fatty acids released from adipocytes flow through the blood to the liver and contribute to hepatic steatosis [6]. This contribution occurs by inducing de novo lipogenesis, resulting from decreased mitochondrial β-oxidation of fat and increasing the synthesis of triglycerides [6]. Waist circumference (WC) reflects the volume of visceral fat better than body mass index (BMI). In addition, because of its cost-effective and simple method of measurement, many current guidelines for the management of obesity recommend measuring WC to assess visceral fat [7,8]. Abdominal obesity (AO), which reflects excessive accumulation of visceral fat, has been considered as one of the major risk factors of NAFLD. Since AO status can change over time, it is more important to consider the long-term trends in AO status rather than spot-checked AO status to determine the risk for metabolic diseases. While there is significant evidence exploring the correlation between different patterns of AO and the incidence of NAFLD in various studies, implications of this relationship may require further investigation and a deeper understanding. Recent studies have highlighted sex differences in fields relevant to NAFLD and its pathogenesis [9].

Therefore, we aimed at assessing the correlation between different AO patterns and the incidence of NAFLD using data from a large community-based, prospective Korean cohort. We also investigated the difference in this influence between male and female patients.

## 2. Materials and Methods

### 2.1. Study Population

We analyzed data from a community-based, prospective study conducted by the Korea Centers for Disease Control and Prevention (KCDC), named the Korean Genome and Epidemiology Study (KoGES)_Ansan_Ansung cohort. A total of 10,030 local residents 40–69 years of age who had lived in urban (Ansan) and rural (Ansung) areas for at least six months were recruited in the baseline survey, which was conducted in 2001–2002, and the participants were requested to participate in the survey biennially until the eighth follow-up (2017–2018). For each follow-up, participants’ medical history, anthropometric measurements, and blood samples were collected.

Participants were followed up from the date of the baseline survey until the time at which the first NAFLD event was ascertained, the end date of the study, or the date of last informative contact. The time from the baseline survey to the first follow-up date was defined as the exposure period. The time from the second to the eighth follow-up date was defined as the event accrual period. We defined new-onset NAFLD as the development of newly diagnosed NAFLD during the event accrual period. The follow-up time was defined as the time interval from the first follow-up date to the time of NAFLD new-onset diagnosis.

Figure 1 shows the flowchart of the study population selection. Among 10,030 participants at baseline, we excluded (1) participants with history of hepatitis (*n* = 423), (2) participants with heavy alcohol consumption ≥30 g/day (for men) or ≥20 g/day (for women) (*n* = 964), (3) those who had insufficient data to calculate NAFLD–liver fat score (*n* = 276), (4) patients with NAFLD in the baseline survey (*n* = 2222), (5) participants without WC data in the baseline survey (*n* = 5), and (6) participants who were never followed up after the baseline survey (*n* = 1546). Of the remaining 5440 participants, we excluded (1) participants without WC measurement on the first follow-up (*n* = 356), (2) those who were never followed up during the event accrual period (*n* = 238), and (3) those who newly developed NAFLD during the first follow-up period (*n* = 379). Finally, we analyzed data from the remaining 4467 participants. The KoGES_Ansan_Ansung cohort protocol was reviewed and approved by the institutional review board (IRB) of the KCDC. Written informed consent was obtained from all participants. This study protocol conformed to the ethical guidelines of the 1964 Declaration of Helsinki and its later amendments. This study was approved by the IRB of Nowon Eulji Medical Center (IRB number: 2021-09-025).

### 2.2. Data Collection

The data on anthropometric measurements such as WC, height, weight, and blood pressure were collected by well-trained examiners. WC (cm) was measured in the horizontal plane, midway between the lowest rib and the iliac crest to the nearest 0.1 cm three times. The average of three WC measurements was used in the analysis. According to the 2018 Korean Society for the Study of Obesity guideline for the management of obesity in Korea [10], AO was defined as WC ≥ 90 cm for men and ≥85 cm for women. Height (cm) and weight (kg) were measured to the nearest 0.1 cm and 0.1 kg, respectively. Body mass index (BMI) (kg/m^2^) was calculated using the height and weight and rounded to the nearest 0.1 kg/m^2^. Systolic (SBP) and diastolic blood pressure (DBP) were measured three times with at least one-minute interval in a sitting position after at least five minutes of rest, and the average of the last two values was calculated. Mean blood pressure (MBP) was also calculated. Body fat percentage and total skeletal muscle mass were obtained from a bioelectrical impedance analysis.

The self-reported questionnaires for dietary habits, smoking status, alcohol consumption, physical activity, monthly household income, and education level were administered. Total energy intake (kcal/day) was calculated using a 103-item food frequency questionnaire. Participants were categorized into never smoker, former smoker, intermittent smoker, and daily smoker. A participant who never smoked or had smoked less than 100 cigarettes in their lifetime was defined as a never smoker. A participant who had smoked more than 100 cigarettes in their lifetime and had quit smoking at the time of the survey was defined as a former smoker. A participant who had smoked more than 100 cigarettes and had not smoked every day during their lifetime was defined as an intermittent smoker. A participant who had smoked more than 100 cigarettes and had smoked every day was defined as a daily smoker. For alcohol consumption, the amount of alcohol intake (g/day) was calculated based on self-reported questionnaire responses using the following equation:The amount of alcohol intake=the average amount of pure alcohol (10 g/per glass of drink)×the number of glasses of alcoholic drink at a time (glasses/time) × the frequency of alcohol use (times/month)÷ 30 (days/month)

Heavy drinkers were defined as those with amount of alcohol intake ≥30 g/day for men and ≥20 g/day for women. The rest of the participants were divided into current drinkers or non-drinkers. Physical activity was measured as metabolic equivalent of task (MET)-hours per week (MET-hr/day) using an International Physical Activity Questionnaire. According to the level of physical activity, participants were categorized into three groups: low, <7.5 MET-hr/day; moderate, 7.5–30 MET-hr/day; and high, >30 MET-hr/day). Monthly household income was categorized into three groups: KRW <100 million, KRW 100–200 million, and KRW >200 million. Education level was categorized into three groups: elementary school or middle school, high school, and college or university.

Blood samples were collected after at least 8 h of fasting. Fasting plasma glucose (FPG), serum glycosylated hemoglobin (HbA1c), insulin, total cholesterol, triglyceride, high-density lipoprotein (HDL) cholesterol, aspartate aminotransferase (AST), alanine aminotransferase (ALT), and C-reactive protein (CRP) levels were measured. Diabetes mellitus (DM) was defined as any of the following: (1) FPG ≥ 126 mg/dL, (2) plasma glucose level ≥ 200 mg/dL at 2 h after the 75-g oral glucose tolerance test, (3) HbA1c ≥ 6.5%, (4) treatment with anti-diabetic agents, or (5) treatment with insulin therapy [11]. Hypertension (HTN) was defined as any of the following: (1) SBP ≥140 mmHg, (2) DBP ≥90 mmHg, or (3) treatment with anti-hypertensive agents [11]. Metabolic syndrome (MetS) was defined as at least three of the following: (1) presence of AO; (2) serum triglyceride level ≥ 150 mg/dL or treatment with lipid-lowering agents; (3) serum HDL cholesterol level < 40 mg/dL (men) or < 50 mg/dL (women); (4) SBP ≥ 130 mmHg, DBP ≥ 85 mmHg, or treatment with anti-hypertensive agents; and (5) FPG level ≥ 100 mg/dL, treatment with anti-diabetic agents, or treatment with insulin therapy [12].

### 2.3. AO Status over Time

We divided participants into four groups according to the change in AO status over the exposure period (baseline to first follow-up period): persistent lean WC, improved AO, progressed to AO, and persistent AO groups. Participants were categorized into (1) persistent lean WC group when they were lean both at baseline and on the first follow-up, (2) improved AO group when they had AO at baseline and were lean on the first follow-up, (3) progressed to AO group when they were lean at baseline and had AO on the first follow-up, and (4) persistent AO group when they had AO both at baseline and on the first follow-up (Appendix A).

### 2.4. Diagnosis of NAFLD

We used the NAFLD–liver fat score to diagnose NAFLD [13], a tool that has undergone validation in the Korean population [14]. The formula for the NAFLD–liver fat score is as follows:NAFLD−liver fat score=−2.89+1.18×MetS (Yes:1,No:0)+0.45×DM (Yes: 2, No: 0)+0.15×insulin in µIU/mL+0.04×AST in U/L−0.94×AST/ALT

We defined NAFLD as NAFLD–liver fat score > −0.640 [13].

### 2.5. Statistical Analysis

We performed all statistical analyses separately for men and women. Data are presented as mean ± standard deviation (SD) for continuous variables and number (percentage) for categorical variables. For continuous variables such as age, WC, body fat percentage, BMI, total skeletal muscle mass, body weight, total energy intake, MBP, FPG, total cholesterol, serum CRP, AST, and ALT levels, analysis of variance was used to compare different groups. For categorical variables, such as smoking status, physical activity, and drinking status, the chi-squared test was used to compare different groups.

Kaplan–Meier curves were used to present the cumulative incidence rate of NAFLD during the event accrual period. We used the log-rank test to determine whether distribution of cumulative incidence rate of NAFLD differed among groups. Hazard ratio (HR) and 95% confidence interval (CI) for the incidence of NAFLD in each AO group were calculated using univariable and multivariable Cox proportional hazards regression analysis. In model 1, we adjusted for sex, age, and BMI. In model 2, we adjusted for variables used in model 1 plus smoking status, alcohol consumption status, physical activity, and total energy intake. In model 3, we adjusted for variables used in model 2 plus MBP, HbA1c, serum total cholesterol, CRP, and ALT levels. To confirm the association between AO pattern and NAFLD incidence, we also examined the association between AO pattern from baseline to second follow-up and NAFLD incidence. All statistical analyses were conducted using SAS version 9.4 (SAS Institute Inc., Cary, NC, USA) and R software (version 4.1.1; R Foundation for Statistical Computing, Vienna, Austria). The significance level was set at *p* < 0.05.

## 3. Results

### 3.1. Baseline Characteristics of the Study Population

The baseline characteristics of the study sample, divided by sex, are listed in Table 1. A total of 4467 participants (1867 men and 2600 women) were included in the final analysis. The mean ± SD of age was 52.1 ± 8.8 years in men and 51.1 ± 8.7 years in women. The mean ± SD of BMI and WC were 23.3 ± 2.6 kg/m^2^ and 80.8 ± 6.7 cm in men and 24.0 ± 2.9 kg/m^2^ and 78.7 ± 8.8 cm in women, respectively.

The baseline characteristics of the study population, divided by sex and NAFLD incidence, are presented in Table 2. In the total population, compared with participants who did not develop NAFLD, participants who newly developed NAFLD had lower HOMA-beta and higher WC, body fat percentage, BMI, skeletal muscle mass, body weight, MBP, FPG, serum total cholesterol, CRP, AST, ALT, insulin levels, and HOMA-IR. The proportions of smoking, physical activity, and current drinkers were not significantly different between the two groups. Men who developed NAFLD were younger and had lower HOMA-beta and higher body fat percentage, BMI, body weight, MBP, FPG, TC, AST, ALT, insulin levels, and HOMA-IR. Women who developed NAFLD were older and had higher WC, body fat percentage, BMI, body weight, MBP, FPG, serum total cholesterol, CRP, AST, insulin levels, and HOMA-IR levels.

### 3.2. Longitudinal Relationship between Longitudinal AO Pattern and NAFLD Incidence

During the median 12.1 years of event accrual period, a total of 1825 (40.86%, 757 men and 1068 women) participants newly developed NAFLD. Figure 2A to Figure 2C present, through Kaplan–Meier curves, the cumulative incidence rate of NAFLD according to the longitudinal AO pattern in the whole population, men, and women. In total population (Figure 2A), the cumulative incidence rate of NAFLD was highest in the persistent AO group, followed by the progressed to AO group, improved AO group, and persistent lean WC group. The persistent AO group had significantly higher NAFLD cumulative incidence rate compared to progressed to AO (log-rank test *p* = 0.002), improved AO (log-rank test *p* < 0.001), and persistent lean WC (log-rank test *p* < 0.001) groups. There was no significant difference in NAFLD cumulative incidence rate between the persistent AO group and progressed to AO group (log-rank test *p* = 0.179). In men (Figure 2B), the persistent AO group had significantly higher NAFLD cumulative incidence rate compared to progressed to the AO (log-rank test *p* = 0.001), improved AO (log-rank test *p* < 0.001), and persistent lean WC (log-rank test *p* < 0.001) groups. There was no significant difference in NAFLD cumulative incidence rate between the persistent AO and improved AO groups (log-rank test *p* = 0.120) or between the persistent AO and progressed to AO groups (log-rank test *p* = 0.467). In women (Figure 2C), a similar trend was observed to that in the total population.

## 4. Discussion

We further confirmed the association between NAFLD and AO pattern by setting the accrual time from baseline to second follow-up time. Similar associations were observed in the whole population, men, and women (Appendix A).

In this study, we found that participants who had persistent AO, progressed to AO, or improved AO had significantly higher risk for NAFLD incidence compared to persistently lean WC participants. Men who had progressed to AO over two years had significantly higher risk for NAFLD than those without any AO. Women who had persistent AO or progressed to AO had significantly higher risk for NAFLD than those who had no AO or improved AO. These associations were noticed even when the observation period was extended for four years.

WC, widely used as a surrogate for AO, has been a well-known risk factor for CVD [15], DM [16], and all-cause mortality [17]. The Multi-Ethnic Study of Atherosclerosis reported that both WC (indicator for AO) and BMI (indicator for general obesity) are important risk factors of NAFLD, but WC better discriminates the risk for NAFLD [18]. In our previous study, we found that WC was the most significant risk factor of incident NAFLD among the various body composition variables (BMI, body fat, and skeletal muscle mass index) in middle-aged and older Korean adults [19]. A recent study conducted in Japan found that AO itself is significantly associated with NAFLD regardless of metabolic health status [20]. These studies have examined the association between baseline WC and NAFLD incidence and had cross-sectional designs.

Several studies have tried to identify the effect of BMI or WC changes on NAFLD risk. A Korean study found a direct relationship between increasing WC and the incidence of NAFLD and an inverse relationship between decreasing WC and the incidence of NAFLD using the domestic single-center cohort study [21]. The authors defined WC changes as quartiles of the difference in WC between baseline and two-year follow-up (Q1, WC loss group; Q3 and Q4, WC gain group) [21]. Although the study considered the WC changes over time, it did not address the effect of the progression to AO or regression from AO on NAFLD incidence. In the current study, the cumulative incidence rate of NAFLD was similar between the persistent AO group and progressed to AO group in both men and women. It is plausible that the impact of changes in AO status observed during the two-year exposure period may gradually diminish or become less pronounced over the extended follow-up period of 14 years. Other factors, such as genetic predisposition, lifestyle changes, or medical interventions, may also come into play and potentially mitigate the initial differences between the two groups. Follow-up studies should be designed to consider various factors to better understand the results.

The Coronary Artery Risk Development in Young Adults study reported that participants with an increase in BMI had greater odds of NAFLD compared to those with stable BMI using the trajectory modeling [22]. Nah et al. [23] found significant association between historical weight changes 10 years ago and the present NAFLD prevalence. They showed that participants who progressed to obesity or overweight had higher risk for NAFLD [23]. The two former studies considered BMI changes using trajectory modeling or pattern changes. However, they did not consider the time by obtaining the relative risk (RR) or odds ratio values. Previous study results are in line with our findings [22,23]. In the current study, we also found that participants who had persistent AO, progressed to AO, or improved AO had higher risk for NAFLD incidence compared to persistently lean participants. Furthermore, we observed the association between WC change pattern and the incidence of NAFLD using the median 12.1-year longitudinal prospective cohort design.

Several possible mechanisms could explain our results. The increase in WC could reflect VAT accumulation [24]. VAT is closely associated with glucose and lipid metabolism. An increase in VAT increases the influx of free fatty acids (FFAs) into the portal circulation and leads to the increase in gluconeogenesis and hepatic insulin resistance (IR), which acts as a key factor for the development of NAFLD [25]. In the state of IR, lipogenesis in the liver increases, and adipose tissue accumulates [26]. Increased hepatic lipogenesis and FFAs lead to disturbed hepatic lipid metabolism and increased lipotoxicity, ultimately resulting in NAFLD [26].

The benefits of weight reduction on histological or biochemical improvement of nonalcoholic steatohepatitis (NASH) and NAFLD are well studied in previous research [27,28]. However, the effect of weight reduction on NAFLD incidence remains unclear. Our findings are in agreement with those of Nah et al. [23], who found that obese or AO participants who subsequently became lean still had higher risk for NAFLD compared to those who were persistently lean (RR: 2.46, 95% CI: 1.40–4.31, *p* = 0.02 in Nah’s study and HR: 1.42, 95% CI: 1.13–1.77, *p* = 0.002 in the current study). Although the exact reasons are unclear, the residual risk related to previous AO history could influence the NAFLD incidence. Further research is needed to investigate the effect of time on fatty liver after regression.

Sex differences were noted in the HR between people who improved AO and those with persistent AO or progression to AO. The HR of men who improved AO was not statistically lower than that of men with persistent AO or progression to AO, while HR of women who improved AO was statistically lower than that of women with persistent AO or progression to AO. For the sex differences in the transition of AO status and the risk for NAFLD, we focused on the differences in sex hormones, percentage of body fat, and smoking status between men and women. Considering a sexual dimorphism in the development of NAFLD [29], estrogen would have a protective role in disease, especially in the absence of AO. However, the protective role of estrogen may be attenuated in the presence of AO in women, which implies that improving AO could be an effective preventive strategy for NAFLD in women. Further research should be performed to confirm the interaction between estrogen and pro-inflammatory cytokines/adipokines from VAT. The different distribution and absolute amount of fat in men and women could also be another explanation for these findings [30]. In this study, we found that women had 1.5 times higher percentage of body fat than men. Women have greater percentage of adipose tissue than men at the same level of BMI; however, women are more likely to store subcutaneous adipose tissue while men are more likely to store VAT for any given amount of fat [31]. In addition, because the cut-off point for defining AO in men is higher than that in women, the absolute amount of VAT could also be higher in men. Compared to women with AO, men with AO may have higher quantities of FFAs, pro-inflammatory cytokines, and adipokines released from VAT, which predisposes them to hepatic steatosis through significant hepatic IR and inflammation. Therefore, the impact of the progression to AO would be more significant in men than in women, as men are likely to experience a greater gain in the absolute amount of VAT. Particularly, men had a strikingly higher proportion of current smokers, almost six times higher than women. The rapid deposition of fat in the liver occurs after smoking [32]. Long-term smoking can also stimulate glucose oxidative metabolism, leading to the suppression of non-oxidative reactions and resulting in elevated levels of plasma FFAs [33]. Therefore, such factors would maintain the risk for developing NAFLD high even if AO improved in men.

Our study has several limitations. First, during the accrual period, it was not possible to track the change in AO status for a relatively short period of time (e.g., short-term diet). We categorized the overall pattern of AO based on only two assessments, one at the beginning and the other at the end of the follow-up period. Second, we defined NAFLD using a surrogate marker (NAFLD–liver fat score) rather than imaging modalities (abdominal ultrasonography or magnetic resonance imaging). The NAFLD–liver fat score lacks the ability to differentiate between mild, moderate, or severe fatty liver or NASH in incident NAFLD. Therefore, this study did not assess the severity of liver fat or fibrosis deposition, which limits our understanding of the clinical implications and disease progression within the NAFLD spectrum. Third, we could not exclude patients with secondary fatty liver diseases such as autoimmune hepatitis, drug-induced hepatitis, or Wilson’s disease. Finally, our study included only middle-aged and older Korean people. Therefore, we could not generalize our findings to other populations or ethnic groups and different age groups. Despite these limitations, our study is the first study to report the association between changes of AO pattern and the incidence of NAFLD using the longitudinal population-based cohort design.

## 5. Conclusions

In total population, persistent AO and progression to AO are associated with higher risk for NAFLD. Persistent AO was a significant risk factor for developing NAFLD only in women, suggesting that both maintaining lean WC or improvement from AO would be effective strategies for preventing NAFLD in women, while maintaining lean WC is more crucial in men. A health strategy that focuses on maintaining a healthy WC throughout life through physical activity and a healthy diet is likely to be more effective in preventing NAFLD than solely relying on reducing WC at a later stage. In addition, considering gender-specific risk profiles can ultimately contribute to the development of more effective health policies and strategies for addressing NAFLD and related health concerns. Additional research is warranted to comprehensively assess the severity of NAFLD in order to obtain a more precise understanding of the relationship between AO and the risk of NAFLD.

## Figures and Tables

**Figure 1 nutrients-15-02880-f001:**
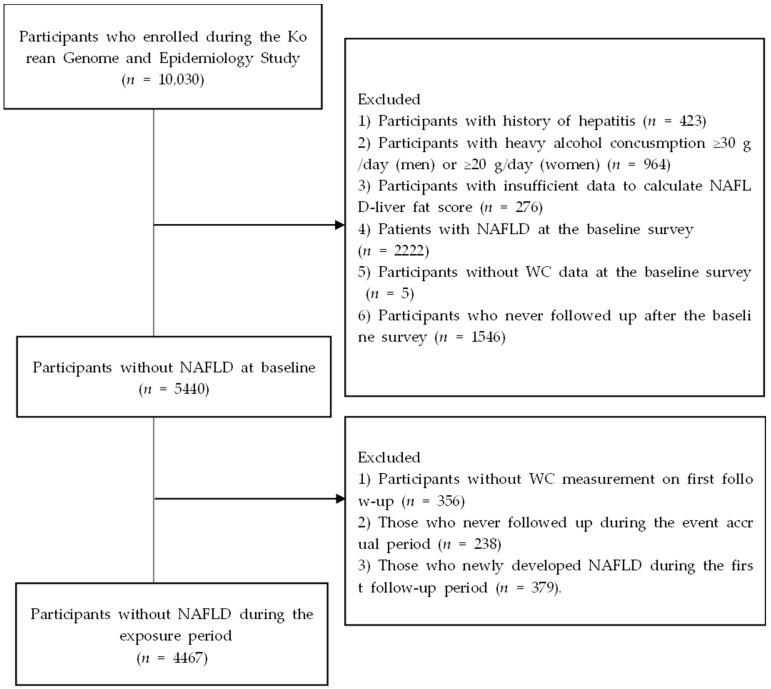
Flow chart of the study population.

**Figure 2 nutrients-15-02880-f002:**
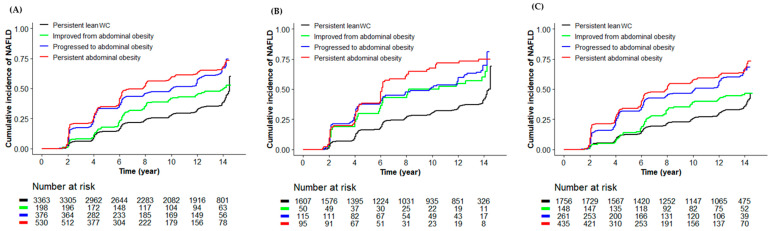
Cumulative incidence of NAFLD according to abdominal obesity pattern: (**A**) total, (**B**) men, and (**C**) women. Table 3 presents the HR (95% CI) for NAFLD incidence according to the longitudinal AO patterns using the Cox proportional hazards regression model. Compared with the reference persistent lean WC group, the HRs (95% CI) for NAFLD incidence in the improved AO, progressed to AO and persistent AO groups were 1.39 (1.13–1.72), 2.26 (1.97–2.61), and 2.56 (2.26–2.89), respectively. In model 3, compared to the persistent lean WC group, the fully adjusted HRs (95% CI) for NAFLD incidence in the improved AO, progressed to AO, and persistent AO groups were 1.06 (0.84–1.33), 1.73 (1.48–2.02), and 1.33 (1.13–1.57), respectively. In pairwise comparison analysis, both the progressed to AO and persistent AO groups had significantly higher risk for developing NAFLD than improved AO group. There was no difference in the risk for developing NAFLD between the progressed to AO and persistent AO groups (Appendix A). In men, compared to the reference persistent lean WC group, the HRs (95% CI) for NAFLD incidence in the improved AO, progressed to AO, and persistent AO groups were 1.91 (1.32–2.77), 2.25 (1.77–2.86), and 2.78 (2.14–3.61), respectively. In the progress to AO group, these significant associations were consistently noticed in models 1, 2, and 3. In pairwise comparison analysis, the progressed to AO group had a significantly higher risk for developing NAFLD than persistent lean WC group. There was no difference in the risk for developing NAFLD between progressed to AO and persistent AO groups or between improved AO and persistent AO groups (Appendix A). In women, compared to the reference persistent lean WC group, the HRs (95% CI) for NAFLD incidence in the improved AO, progressed to AO, and persistent AO groups were 1.33 (1.03–1.73), 2.38 (2.00–2.84), and 2.70 (2.33–3.12), respectively. In the progress to AO and persistent AO groups, these associations remained statistically significant in adjusted models. In pairwise comparison analysis, women showed similar patterns of association to those in the total population (Appendix A).

**Table 1 nutrients-15-02880-t001:** Baseline characteristics of the study population.

Variables	Total (*n* = 4467)	Men (*n* = 1867)	Women (*n* = 2600)	*p*-Value
Age, years	51.6 ± 8.8	52.1 ± 8.8	51.1 ± 8.7	<0.0001
WC, cm	79.6 ± 7.9	80.8 ± 6.7	78.7 ± 8.5	<0.0001
Body fat, %	26.0 ± 7.1	20.1 ± 4.9	30.3 ± 5.3	<0.0001
BMI, kg/m^2^	23.7 ± 2.8	23.3 ± 2.6	24.0 ± 2.9	<0.0001
Body weight, kg	60.1 ± 8.9	64.6 ± 8.6	56.9 ± 7.7	<0.0001
Smoking status, n (%)				<0.0001
Non-smoker	2978 (67.5)	501 (27.0)	2477 (96.9)	
Ex-smoker	562 (12.7)	545 (29.3)	17 (0.7)	
Intermittent smoker	81 (1.8)	67 (3.6)	14 (0.6)	
Daily smoker	793 (18.0)	745 (40.1)	48 (1.9)	
Physical activity, n (%)				<0.0001
Low	306 (7.1)	98 (5.5)	208 (8.3)	
Moderate	2601 (60.5)	1017 (56.8)	1584 (63.2)	
High	1389 (32.3)	675 (37.7)	714 (28.5)	
Currently drinking, n (%)	1907 (43.0)	1198 (64.6)	709 (27.5)	<0.0001
Total energy intake, kcal/day	1934.1 ± 705.5	1991.6 ± 671.3	1893.0 ± 726.3	<0.0001
Mean blood pressure, mmHg	93.6 ± 12.4	95.1 ± 11.7	92.6 ± 12.8	<0.0001
Fasting glucose, mg/dL	82.5 ± 12.2	84.3 ± 13.3	81.2 ± 11.2	<0.0001
Total cholesterol, mg/dL	187.5 ± 33.3	189.2 ± 33.3	186.3 ± 33.2	0.004
hsCRP, mg/dL	0.23 ± 0.63	0.24 ± 0.52	0.22 ± 0.70	0.437
ALT, IU/L	21.9 ± 8.70	25.6 ± 9.8	19.2 ± 6.6	<0.0001
AST, IU/L	26.3 ± 6.8	28.0 ± 7.2	25.2 ± 6.2	<0.0001
HOMA-IR	1.320 ± 0.595	1.237 ± 0.567	1.379 ± 0.607	<0.0001
HOMA-beta	151.392 ± 136.726	127.824 ± 125.483	168.303 ± 141.886	<0.0001
Insulin	6.479 ± 2.772	5.944 ± 2.588	6.864 ± 2.836	<0.0001

Abbreviations: ALT, alanine aminotransferase; AST, aspartate aminotransferase; BMI, body mass index; hsCRP, high-sensitivity C-reactive protein; WC, waist circumference; HOMA-IR, homeostatic model assessment for insulin resistance.

**Table 2 nutrients-15-02880-t002:** Baseline characteristics of men and women according to incident non-alcoholic fatty liver disease.

	Total			Men			Women		
	No Incident NAFLD (*n* = 2642)	Incident NAFLD (*n* = 1825)	*p*-Value	No Incident NAFLD (*n* = 1110)	Incident NAFLD (*n* = 757)	*p*-Value	No Incident NAFLD (*n* = 1532)	Incident NAFLD (*n* = 1068)	*p*-Value
Age, years	51.4 ± 9.0	51.8 ± 8.4	0.124	52.6 ± 9.1	51.5 ± 8.3	0.010	50.5 ± 8.8	52.0 ± 8.5	<0.0001
WC, cm	77.6 ± 7.7	82.5 ± 7.3	<0.001	79.0 ± 6.6	83.4 ± 6.0	<0.001	76.6 ± 8.2	81.8 ± 8.0	<0.0001
Body fat, %	24.9 ± 7.1	27.7 ± 6.9	<0.001	19.1 ± 4.8	21.6 ± 5.0	<0.001	29.1 ± 5.2	32.0 ± 4.8	<0.0001
BMI, kg/m^2^	23.0 ± 2.6	24.7 ± 2.7	<0.001	22.7 ± 2.5	24.2 ± 2.4	<0.001	23.2 ± 2.7	25.1 ± 2.8	<0.0001
Weight, kg	58.4 ± 8.5	62.6 ± 8.9	<0.001	62.8 ± 8.2	67.3 ± 8.3	<0.001	55.3 ± 7.3	59.2 ± 7.8	<0.0001
Smoking status, n (%)			0.817			0.180			0.231
Non-smoker	1777 (68.0)	1201 (66.8)		317 (28.7)	184 (24.4)		1460 (96.6)	1017 (97.4)	
Ex-smoker	332 (12.7)	230 (12.8)		322 (29.2)	223 (29.5)		10 (0.7)	7 (0.7)	
Intermittent smoker	46 (1.8)	35 (2.0)		39 (3.5)	28 (3.7)		7 (0.5)	7 (0.7)	
Daily smoker	460 (17.6)	333 (18.5)		425 (38.5)	320 (42.4)		35 (2.3)	13 (1.3)	
Physical activity, n (%)			0.778			0.889			0.401
Low	186 (7.3)	120 (6.9)		58 (5.4)	40 (5.6)		128 (8.7)	80 (7.8)	
Moderate	1544 (60.7)	1057 (60.4)		603 (56.4)	414 (57.4)		941 (63.8)	643 (62.4)	
High	815 (32.0)	574 (32.8)		408 (38.2)	267 (37.0)		407 (27.6)	307 (29.8)	
Currently drinking, n (%)	1103 (42.1)	804 (44.4)	0.129	698 (63.4)	500 (66.3)	0.197	405 (26.7)	304 (28.8)	0.241
Total energy intake, kcal/day	1927.8 ± 681.0	1943.3 ± 739.6	0.486	1978.9 ± 654.8	2010.1 ± 694.7	0.331	1891.1 ± 697.1	1895.6 ± 766.9	0.880
Mean blood pressure, mmHg	92.1 ± 12.4	95.7 ± 12.2	<0.001	94.1 ± 11.8	96.6 ± 11.4	<0.001	90.6 ± 12.6	95.1 ± 12.7	<0.001
Fasting glucose, mg/dL	80.9 ± 8.4	84.7 ± 15.9	<0.001	82.4 ± 9.4	87.0 ± 17.1	<0.001	79.9 ± 7.4	83.0 ± 14.8	<0.001
Total cholesterol, mg/dL	185.0 ± 32.7	191.2 ± 33.8	<0.001	186.7 ± 33.1	192.8 ± 33.5	0.001	183.7 ± 32.3	190.0 ± 34.1	<0.001
hsCRP, mg/dL	0.21 ± 0.50	0.25 ± 0.78	0.024	0.23 ± 0.54	0.24 ± 0.49	0.536	0.19 ± 0.46	0.26 ± 0.93	0.025
ALT, IU/L	20.9 ± 8.1	23.3 ± 9.3	<0.001	24.1 ± 9.2	27.7 ± 10.4	<0.001	18.6 ± 6.4	20.2 ± 6.9	<0.001
AST, IU/L	26.1 ± 6.6	26.6 ± 7.1	0.016	27.7 ± 7.0	28.5 ± 7.5	0.018	25.0 ± 6.0	25.3 ± 6.4	0.214
HOMA-IR	1.261 ± 0.561	1.405 ± 0.631	<0.001	1.177 ± 0.533	1.326 ± 0.603	<0.001	1.321 ± 0.572	1.462 ± 0.645	<0.0001
HOMA-beta	155.186 ± 137.717	145.890 ± 135.126	0.026	133.450 ± 144.596	119.553 ± 89.774	0.011	170.934 ± 130.311	164.525 ± 157.008	0.273
Insulin	6.288 ± 2.677	6.757 ± 2.883	<0.001	5.759 ± 2.496	6.217 ± 2.697	<0.001	6.671 ± 2.738	7.140 ± 2.949	<0.0001

Abbreviations: ALT, alanine aminotransferase; AST, aspartate aminotransferase; BMI, body mass index; hsCRP, high-sensitivity C-reactive protein; WC, waist circumference; HOMA-IR, homeostatic model assessment for insulin resistance.

**Table 3 nutrients-15-02880-t003:** Cox proportional hazards regression analysis for incident non-alcoholic fatty liver disease according to abdominal obesity patterns.

		Unadjusted		Model 1		Model 2		Model 3	
		HR (95% CI)	*p*-Value	HR (95% CI)	*p*-Value	HR (95% CI)	*p*-Value	HR (95% CI)	*p*-Value
**Total**	Persistent lean WC	ref		ref		ref		ref	
	Improved abdominal obesity	1.39 (1.13–1.72)	0.002	1.03 (0.83–1.27)	0.822	1.02 (0.81–1.28)	0.852	1.06 (0.84–1.33)	0.637
	Progressed to abdominal obesity	2.26 (1.97–2.61)	<0.001	1.73 (1.52–2.03)	<0.001	1.77 (1.52–2.06)	<0.001	1.73 (1.48–2.02)	<0.001
	Persistent abdominal obesity	2.56 (2.26–2.89)	<0.001	1.32 (1.13–1.54)	<0.001	1.29 (1.09–1.52)	0.003	1.33 (1.13–1.57)	<0.001
**Men**	Persistent lean WC	ref		ref		ref		ref	
	Improved abdominal obesity	1.91 (1.32–2.77)	0.001	1.30 (0.89–1.90)	0.180	1.34 (0.90–1.98)	0.146	1.47 (0.99–2.18)	0.055
	Progressed to abdominal obesity	2.25 (1.77–2.86)	<0.001	1.60 (1.25–2.05)	<0.001	1.57 (1.21–2.05)	<0.001	1.60 (1.22–2.09)	<0.001
	Persistent abdominal obesity	2.78 (2.14–3.61)	<0.001	1.20 (0.88–1.64)	0.249	1.12 (0.81–1.54)	0.505	1.21 (0.87–1.69)	0.253
**Women**	Persistent lean WC	ref		ref		ref		ref	
	Improved abdominal obesity	1.33 (1.03–1.73)	0.029	0.95 (0.73–1. 23)	0.691	0.93 (0.70–1.23)	0.586	0.93 (0.71–1.24)	0.628
	Progressed to abdominal obesity	2.38 (2.00–2.84)	<0.001	1.81 (1.51–2.17)	<0.001	1.83 (1.51–2.21)	<0.001	1.78 (1.47–2.16)	<0.001
	Persistent abdominal obesity	2.70 (2.33–3.12)	<0.001	1.31 (1.09–1.58)	0.004	1.31 (1.07–1.59)	0.008	1.36 (1.12–1.65)	0.002

Model 1: adjusted for age, sex (in total population), and body mass index; model 2: model 1 plus smoking, physical activity, alcohol consumption, and total energy intake; model 3: model 2 plus mean blood pressure, glycosylated hemoglobin, total cholesterol, high-sensitivity C-reactive protein, and alanine aminotransferase.

## Data Availability

The dataset used in this study can be provided after review and evaluation of the research plan by the Korea Centers for Disease Control and Prevention (http://www.cdc.go.kr/CDC/eng/main.jsp).

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
