# Peer review of "Gender Differences in the Risk for Incident Non-Alcoholic Fatty Liver Disease According to the Transition of Abdominal Obesity Status: A 16-Year Cohort Study"

_nutrients, 2023, doi:10.3390/nu15132880_

Round 1
Reviewer 1 Report
In this study, the authors report the association between changes of AO pattern and the incidence of NAFLD using the longitudinal population-based cohort design in Korean men and women.
Please address the following comments
1-Did the authors identify alcohol consumption based on participants filled questionnaires or blood/ urine assessments?
2-can the authors comment on the difference in body fat and smoking status between men and women as it is almost 1.5 times and 6 times less, respectively?
3- Table 2 formatting is required, it is currently hard to draw conclusions about correlations between physical activity and NAFLD incidence
4- is there an interpretation for the difference between men (fig 2B) and women (fig 3C), persistent >progressed (men) while persistent =progressed (women). Also, by 14 month, the lean men start to develop NAFLD and progressed has high incidence than persistent
5-table3, among all these factor (smoking, physical activity, alcohol drinking, total energy intake. Model 243 3: Model 2 plus smoking, physical activity, alcohol drinking, total energy intake, mean blood pressure, fasting glucose, total cholesterol, hsCRP and ALT) appears to have the highest incidence of NAFLD? IS IT different between men and women?
6- the disturbution of fat differs between men and women where in men it is mainly abdominal while in women it is mainly subcutaneously, how can the authors interpret this in the light of their findings?
7-In the discussion line 279, Previous study results are in line with our findings. Add ref
8- would the authors add the translational impact of this study?
9- please consider grammatical and language check

can be improved
Author Response
Response to reviewer #1.
In this study, the authors report the association between changes of AO pattern and the incidence of NAFLD using the longitudinal population-based cohort design in Korean men and women.
Please address the following comments
1-Did the authors identify alcohol consumption based on participants filled questionnaires or blood/ urine assessments?
Response: Thank you for the question. Information on alcohol consumption was assessed based on the questionnaire. We have included the detailed description on the assessment of alcohol consumption in the revised Materials and Methods section.
|
2. Materials and Methods 2.2. Data collection For alcohol consumption, the amount of alcohol intake (g/day) was calculated based on self-reported questionnaire responses using the following equation: The amount of alcohol intake = the average amount of pure alcohol (10 g/per glass of drink) × the number of glasses of alcoholic drink at a time (glasses/time) × the frequency of alcohol use (times/month) ÷ 30 (days/month). Heavy drinkers were defined as those with amount of alcohol intake ≥30 g/day for men and ≥20 g/day for women. The rest of the participants were divided into current drinkers or non-drinkers. |
2-can the authors comment on the difference in body fat and smoking status between men and women as it is almost 1.5 times and 6 times less, respectively?
Response: We appreciate the valuable suggestion from reviewer #1. Upon considering the differences in percentage of body fat and smoking status between men and women, we believe that these factors could provide valuable clues to explain our results. We have now incorporated this point into the revised Discussion section.
|
4. Discussion Sex differences were noted in the HR between people who improved AO and those with persistent AO or progression to AO. The HR of men who improved AO was not statistically lower than that of men with persistent AO or progression to AO, while HR of women who improved AO was statistically lower than that of women with persistent AO or progression to AO. For the sex differences in the transition of AO status and the risk for NAFLD, we focus on the differences in sex hormones, percentage of body fat, and smoking status between men and women. Considering a sexual dimorphism in the development of NAFLD [29], estrogen would have a protective role in disease, especially in the absence of AO. However, the protective role of estrogen may be attenuated in the presence of AO in women, which implies that improving AO could be an effective preventive strategy for NAFLD in women. Further research should be performed to confirm the interaction between estrogen and pro-inflammatory cytokines/adipokines from VAT. The different distribution and absolute amount of fat in men and women could also be another explanation for these findings [30]. In this study, we found that women had 1.5 times higher percentage of body fat than men. Women have greater percentage of adipose tissue than men at the same level of BMI; however, women are more likely to store subcutaneous adipose tissue while men are more likely to store VAT for any given amount of fat [31]. In addition, because the cut-off point for defining AO in men is higher than that in women, the absolute amount of VAT could also be higher in men. Compared to women with AO, men with AO may have higher amounts of FFA, pro-inflammatory cytokines, and adipokines released from VAT, which predisposes them to hepatic steatosis through significant hepatic IR and inflammation. Therefore, the impact of the progression to AO would be more significant in men than in women, as men are likely to experience a greater gain in the absolute amount of VAT. Particularly, men had strikingly higher proportion of current smokers, almost six times higher than women. The rapid deposition of fat in the liver occurs after smoking [32]. Long-term smoking can also stimulate glucose oxidative metabolism, leading to the suppression of non-oxidative reactions and resulting in elevated levels of plasma FFA [33]. Therefore, such factors would maintain the risk for developing NAFLD high even if AO improved in men.
References 32. Liu, Y.; Dai, M.; Bi, Y.; Xu, M.; Xu, Y.; Li, M.; Wang, T.; Huang, F.; Xu, B.; Zhang, J.; et al. Active smoking, passive smoking, and risk of nonalcoholic fatty liver disease (NAFLD): a population-based study in China. J Epidemiol 2013, 23, 115-121, doi:10.2188/jea.je20120067. 33. Zhang, C.X.; Guo, L.K.; Qin, Y.M.; Li, G.Y. Association of polymorphisms of adiponectin gene promoter-11377C/G, glutathione peroxidase-1 gene C594T, and cigarette smoking in nonalcoholic fatty liver disease. J Chin Med Assoc 2016, 79, 195-204, doi:10.1016/j.jcma.2015.09.003. |
3- Table 2 formatting is required, it is currently hard to draw conclusions about correlations between physical activity and NAFLD incidence
Response: We apologize for the lack of clarity in presenting Table 2. We have edited Table 2 to improve clarity.
Table 2. Baseline characteristics of men and women according to incident non-alcoholic fatty liver disease.
|
|
Total |
|
|
Men |
|
|
Women |
|
|
|
|
No incident NAFLD (n=2642) |
Incident NAFLD (n=1825) |
P-value |
No incident NAFLD (n=1,110) |
Incident NAFLD (n=757) |
P-value |
No incident NAFLD (n=1,532) |
Incident NAFLD (n=1,068) |
P-value |
|
Age, years |
51.4 ± 9.0 |
51.8 ± 8.4 |
0.124 |
52.6 ± 9.1 |
51.5 ± 8.3 |
0.010 |
50.5 ± 8.8 |
52.0 ± 8.5 |
<0.001 |
|
WC, cm |
77.6 ± 7.7 |
82.5 ± 7.3 |
<.001 |
79.0 ± 6.6 |
83.4 ± 6.0 |
<.001 |
76.6 ± 8.2 |
81.8 ± 8.0 |
<0.001 |
|
Body fat, % |
24.9 ± 7.1 |
27.7 ± 6.9 |
<.001 |
19.1 ± 4.8 |
21.6 ± 5.0 |
<.001 |
29.1 ± 5.2 |
32.0 ± 4.8 |
<0.001 |
|
BMI, kg/m2 |
23.0 ± 2.6 |
24.7 ± 2.7 |
<.001 |
22.7 ± 2.5 |
24.2 ± 2.4 |
<.001 |
23.2 ± 2.7 |
25.1 ± 2.8 |
<0.001 |
|
Weight, kg |
58.4 ± 8.5 |
62.6 ± 8.9 |
<.001 |
62.8 ± 8.2 |
67.3 ± 8.3 |
<.001 |
55.3 ± 7.3 |
59.2 ± 7.8 |
<0.001 |
|
Smoking status, n (%) |
|
|
0.817 |
|
|
0.180 |
|
|
0.231 |
|
Non-smoker |
1777 (68.0) |
1201 (66.8) |
|
317 (28.7) |
184 (24.4) |
|
1460 (96.6) |
1017 (97.4) |
|
|
Ex-smoker |
332 (12.7) |
230 (12.8) |
|
322 (29.2) |
223 (29.5) |
|
10 (0.7) |
7 (0.7) |
|
|
Intermittent smoker |
46 (1.8) |
35 (2.0) |
|
39 (3.5) |
28 (3.7) |
|
7 (0.5) |
7 (0.7) |
|
|
Daily smoker |
460 (17.6) |
333 (18.5) |
|
425 (38.5) |
320 (42.4) |
|
35 (2.3) |
13 (1.3) |
|
|
Physical activity, n (%) |
|
|
0.778 |
|
|
0.889 |
|
|
0.401 |
|
Low |
186 (7.3) |
120 (6.9) |
|
58 (5.4) |
40 (5.6) |
|
128 (8.7) |
80 (7.8) |
|
|
Moderate |
1544 (60.7) |
1057 (60.4) |
|
603 (56.4) |
414 (57.4) |
|
941 (63.8) |
643 (62.4) |
|
|
High |
815 (32.0) |
574 (32.8) |
|
408 (38.2) |
267 (37.0) |
|
407 (27.6) |
307 (29.8) |
|
|
Currently drinking, n (%) |
1103 (42.1) |
804 (44.4) |
0.129 |
698 (63.4) |
500 (66.3) |
0.197 |
405 (26.7) |
304 (28.8) |
0.241 |
|
Total energy intake, kcal/day |
1927.8 ± 681.0 |
1943.3 ± 739.6 |
0.486 |
1978.9 ± 654.8 |
2010.1 ± 694.7 |
0.331 |
1891.1 ± 697.1 |
1895.6 ± 766.9 |
0.880 |
|
Mean blood pressure, mmHg |
92.1 ± 12.4 |
95.7 ± 12.2 |
<.001 |
94.1 ± 11.8 |
96.6 ± 11.4 |
<.001 |
90.6 ± 12.6 |
95.1 ± 12.7 |
<.001 |
|
Fasting glucose, mg/dL |
80.9 ± 8.4 |
84.7 ± 15.9 |
<.001 |
82.4 ± 9.4 |
87.0 ± 17.1 |
<.001 |
79.9 ± 7.4 |
83.0 ± 14.8 |
<.001 |
|
Total cholesterol, mg/dL |
185.0 ± 32.7 |
191.2 ± 33.8 |
<.001 |
186.7 ± 33.1 |
192.8 ± 33.5 |
0.001 |
183.7 ± 32.3 |
190.0 ± 34.1 |
<.001 |
|
hsCRP, mg/dL |
0.21 ± 0.50 |
0.25 ± 0.78 |
0.024 |
0.23 ± 0.54 |
0.24 ± 0.49 |
0.536 |
0.19 ± 0.46 |
0.26 ± 0.93 |
0.025 |
|
ALT, IU/L |
20.9 ± 8.1 |
23.3 ± 9.3 |
<.001 |
24.1 ± 9.2 |
27.7 ± 10.4 |
<.001 |
18.6 ± 6.4 |
20.2 ± 6.9 |
<.001 |
|
AST, IU/L |
26.1 ± 6.6 |
26.6 ± 7.1 |
0.016 |
27.7 ± 7.0 |
28.5 ± 7.5 |
0.018 |
25.0 ± 6.0 |
25.3 ± 6.4 |
0.214 |
Abbreviations: ALT, alanine aminotransferase; AST, aspartate aminotransferase; BMI, body mass index; hsCRP, high-sensitivity C-reactive protein; WC, waist circumference
4- is there an interpretation for the difference between men (fig 2B) and women (fig 3C), persistent >progressed (men) while persistent =progressed (women). Also, by 14 month, the lean men start to develop NAFLD and progressed has high incidence than persistent
Response: Thank you for the comment. Cumulative incidence rate of NAFLD seems to be higher in progress to AO group than in persistent AO group in men and similar between persistent and progress to AO groups in women. It is plausible that the impact of changes in AO status observed during the two-year exposure period may gradually diminish or become less pronounced over the extended follow-up period of 14 years. Other factors, such as genetic predisposition, lifestyle changes, or medical interventions, may also come into play and potentially mitigate the initial differences between the two groups. We have included this point in the revised Discussion section. The high incidence of the persistent lean group progressing to the AO group in men observed at 14 years may indeed be related to censored data. Kaplan–Meier curves are commonly used in survival analysis, where censored data represents cases where the event (e.g., death or disease occurrence) has not yet happened or the observation is incomplete until the end of the study period. Censored data occurs when the exact survival time is unknown, for example, when the study ends or when a patient is lost to follow-up during the study. Therefore, when there is a sudden increase in the incidence rate at the last observed time point, it may indicate that some of the previously censored cases experienced the event. This increase can occur because of the presence of more censored data at the endpoint.
|
4. Discussion Several studies have tried to identify the effect of BMI or WC changes on NAFLD risk. A Korean study found a direct relationship between increasing WC and the incidence of NAFLD and an inverse relationship between decreasing WC and the incidence of NAFLD using the domestic single-center cohort study [21]. The authors defined WC changes as quartiles of the difference in WC between baseline and two-year follow-up (Q1, WC loss group; Q3 and Q4, WC gain group) [21]. Although the study considered the WC changes over time, it did not address the effect of the progression to AO or regression from AO on NAFLD incidence. In the current study, the cumulative incidence rate of NAFLD was similar between persistent AO group and progressed to AO group in both men and women. It is plausible that the impact of changes in AO status observed during the two-year exposure period may gradually diminish or become less pronounced over the extended follow-up period of 14 years. Other factors, such as genetic predisposition, lifestyle changes, or medical interventions, may also come into play and potentially mitigate the initial differences between the two groups. Follow-up studies should be designed to consider various factors to better understand the results. |
5-table3, among all these factor (smoking, physical activity, alcohol drinking, total energy intake. Model 243 3: Model 2 plus smoking, physical activity, alcohol drinking, total energy intake, mean blood pressure, fasting glucose, total cholesterol, hsCRP and ALT) appears to have the highest incidence of NAFLD? IS IT different between men and women?
Response: Thank you for your comment. Table 3 provides information on various factors, including smoking, physical activity, alcohol drinking, total energy intake, and different models used to assess the incidence of NAFLD. We presented the difference between men and women in the Table 1. Although there was no difference in mean serum hsCRP level between men and women, we adjusted for this variable considering its clinical importance.
Table 1. Baseline characteristics of the study population.
|
Variables |
Total (n=4,467) |
Men (n=1,867) |
Women (n=2,600) |
P-value |
|
Age, years |
51.6 ± 8.8 |
52.1 ± 8.8 |
51.1 ± 8.7 |
<0.001 |
|
WC, cm |
79.6 ± 7.9 |
80.8 ± 6.7 |
78.7 ± 8.5 |
<0.001 |
|
Body fat, % |
26.0 ± 7.1 |
20.1 ± 4.9 |
30.3 ± 5.3 |
<0.001 |
|
BMI, kg/m2 |
23.7 ± 2.8 |
23.3 ± 2.6 |
24.0 ± 2.9 |
<0.001 |
|
Body weight, kg |
60.1 ± 8.9 |
64.6 ± 8.6 |
56.9 ± 7.7 |
<0.001 |
|
Smoking status, n (%) |
|
|
|
<0.001 |
|
Non-smoker |
2978 (67.5) |
501 (27.0) |
2477 (96.9) |
|
|
Ex-smoker |
562 (12.7) |
545 (29.3) |
17 (0.7) |
|
|
Intermittent smoker |
81 (1.8) |
67 (3.6) |
14 (0.6) |
|
|
Daily smoker |
793 (18.0) |
745 (40.1) |
48 (1.9) |
|
|
Physical activity, n (%) |
|
|
|
<0.001 |
|
Low |
306 (7.1) |
98 (5.5) |
208 (8.3) |
|
|
Moderate |
2601 (60.5) |
1017 (56.8) |
1584 (63.2) |
|
|
High |
1389 (32.3) |
675 (37.7) |
714 (28.5) |
|
|
Currently drinking, n (%) |
1907 (43.0) |
1198 (64.6) |
709 (27.5) |
<0.001 |
|
Total energy intake, kcal/day |
1934.1 ± 705.5 |
1991.6 ± 671.3 |
1893.0 ± 726.3 |
<0.001 |
|
Mean blood pressure, mmHg |
93.6 ± 12.4 |
95.1 ± 11.7 |
92.6 ± 12.8 |
<0.001 |
|
Fasting glucose, mg/dL |
82.5 ± 12.2 |
84.3 ± 13.3 |
81.2 ± 11.2 |
<0.001 |
|
Total cholesterol, mg/dL |
187.5 ± 33.3 |
189.2 ± 33.3 |
186.3 ± 33.2 |
0.004 |
|
hsCRP, mg/dL |
0.23 ± 0.63 |
0.24 ± 0.52 |
0.22 ± 0.70 |
0.437 |
|
ALT, IU/L |
21.9 ± 8.70 |
25.6 ± 9.8 |
19.2 ± 6.6 |
<0.001 |
|
AST, IU/L |
26.3 ± 6.8 |
28.0 ± 7.2 |
25.2 ± 6.2 |
<0.001 |
|
HOMA-IR |
1.320 ± 0.595 |
1.237 ± 0.567 |
1.379 ± 0.607 |
<0.001 |
|
HOMA-beta |
151.392 ± 136.726 |
127.824 ± 125.483 |
168.303 ± 141.886 |
<0.001 |
|
Insulin |
6.479 ± 2.772 |
5.944 ± 2.588 |
6.864 ± 2.836 |
<0.001 |
Abbreviations: ALT, alanine aminotransferase; AST, aspartate aminotransferase; BMI, body mass index; hsCRP, high-sensitivity C-reactive protein; WC, waist circumference; HOMA-IR, homeostatic model assessment for insulin resistance.
6- the disturbution of fat differs between men and women where in men it is mainly abdominal while in women it is mainly subcutaneously, how can the authors interpret this in the light of their findings?
Response: We completely agree with reviewer #1's point of view. We believe that the different distribution of body fat can be one of the reasons for the disparate results observed between men and women. We have discussed this point in the Discussion section and provided further interpretation regarding its implications.
|
4. Discussion Sex differences were noted in the HR between people who improved AO and those with persistent AO or progression to AO. The HR of men who improved AO was not statistically lower than that of men with persistent AO or progression to AO, while HR of women who improved AO was statistically lower than that of women with persistent AO or progression to AO. For the sex differences in the transition of AO status and the risk for NAFLD, we focus on the differences in sex hormones, percentage of body fat, and smoking status between men and women. Considering a sexual dimorphism in the development of NAFLD [29], estrogen would have a protective role in disease, especially in the absence of AO. However, the protective role of estrogen may be attenuated in the presence of AO in women, which implies that improving AO could be an effective preventive strategy for NAFLD in women. Further research should be performed to confirm the interaction between estrogen and pro-inflammatory cytokines/adipokines from VAT. The different distribution and absolute amount of fat in men and women could also be another explanation for these findings [30]. In this study, we found that women had 1.5 times higher percentage of body fat than men. Women have greater percentage of adipose tissue than men at the same level of BMI; however, women are more likely to store subcutaneous adipose tissue while men are more likely to store VAT for any given amount of fat [31]. In addition, because the cut-off point for defining AO in men is higher than that in women, the absolute amount of VAT could also be higher in men. Compared to women with AO, men with AO may have higher amounts of FFA, pro-inflammatory cytokines, and adipokines released from VAT, which predisposes them to hepatic steatosis through significant hepatic IR and inflammation. Therefore, the impact of the progression to AO would be more significant in men than in women, as men are likely to experience a greater gain in the absolute amount of VAT. |
7-In the discussion line 279, Previous study results are in line with our findings. Add ref
Response: We have added references for the sentence in the revised Discussion section.
|
4. Discussion Previous study results are in line with our findings [22,23]. |
8- would the authors add the translational impact of this study?
Response: Thank you for your comment. We added the following sentence in the conclusion section.
|
Line 378 In total population, persistent AO and progression to AO are associated with higher risk for NAFLD. Persistent AO was a significant risk factor for developing NAFLD only in women, suggesting that both maintaining lean WC or improvement from AO would be effective strategies for preventing NAFLD in women while maintaining lean WC is more crucial in men. A health strategy that focuses on maintaining a healthy WC throughout life through physical activity and a healthy diet is likely to be more effective in preventing NAFLD than solely relying on reducing WC at a later stage. Also, considering gender-specific risk profiles can ultimately contribute to the development of more effective health policies and strategies for addressing NAFLD and related health concerns. Additional research is warranted to comprehensively assess the severity of NAFLD, in order to obtain a more precise understanding of the relationship between AO and the risk of NAFLD. |
9- please consider grammatical and language check
Response: Thank you for your comment. The manuscript and submission files, written by authors for whom English is a second language, have been edited by an editor specializing in editing scientific manuscripts.

Reviewer 2 Report
This paper is an interesting paper presenting the association between the transition of WC and incident NAFLD. However, I would only call the conclusion ‘preliminary’ due to the following reasons:
1) the validity of disease definition. the NAFLD-liver fat score was developed in a small cross-sectional population with ‘The optimal cut-off point of −0.640 predicted increased liver fat content with a sensitivity of 86% and specificity of 71%’. Whether this score can be used to predict ‘incident’ NAFLD in a Korean population-based cohort is invalid. How accurate is this score to predict incident NAFLD in Korean cohorts? Please find previous scores to support choosing this score.
2)Another issue is that NAFLD-liver fat score is unable to tell the severity of liver fat or fibrosis deposition in the liver. It is unable to tell whether the incident NAFLD is a mild, moderate or severe fatty liver or NASH. The clinical significance of NAFLD was not as much as incident Diabetes, which was used to calculate the NAFLD-liver fat score. If the authors were unable to find imaging data to define NAFLD, then a proper discussion of limitations and a mild tone in the conclusion should be used.
- the authors have excluded more than 2000 participants with previous NAFLD, are they defined by ultrasound or NAFLD-liver fat score? Please specify. The authors has excluded more than half of the total cohort (10,030), sensitivity analysis should be expended.
Minor issues are:
1)Please present the comparison of clinical characteristics among the four groups (Persistent lean, Improved from abdominal obesity, Progressed to abdominal obesity, Persistent abdominal obesity).
- please compare HOMA-IR and HOMA-B, fasting glucose, and Insulin in all tables.
- current table 3: Please change covariate FPG to HbA1c if it is possible. Try to adjust BMI if possible.
- I did not see Figure 1 in the manuscript.
- The name of the four groups is a bit misleading, persistent lean indicates normal BMI rather than WC.
Author Response
Response to reviewer #2.
This paper is an interesting paper presenting the association between the transition of WC and incident NAFLD. However, I would only call the conclusion ‘preliminary’ due to the following reasons:
1) the validity of disease definition. the NAFLD-liver fat score was developed in a small cross-sectional population with ‘The optimal cut-off point of −0.640 predicted increased liver fat content with a sensitivity of 86% and specificity of 71%’. Whether this score can be used to predict ‘incident’ NAFLD in a Korean population-based cohort is invalid. How accurate is this score to predict incident NAFLD in Korean cohorts? Please find previous scores to support choosing this score.
Response: We appreciate reviewer #2’s insightful comment. A study conducted in 2014 validated the NAFLD-liver fat score for the Korean population. In the development dataset comprising 15,676 participants, the area under the receiver operating characteristic curve (AUC) of the NAFLD-liver fat score was 0.7782 (sensitivity 37%, specificity 94%, positive predictive value 81%, and negative predictive value 68%). In the external validation dataset, which included a total of 66,868 participants, the AUC of the NAFLD-liver fat score was 0.82 (sensitivity 40%, specificity 94%, positive predictive value 75%, and negative predictive value 79%). We have included this reference in the Materials and Methods section of the revised manuscript. Additionally, we performed a sensitivity analysis, defining NAFLD using the hepatic steatosis index, which was developed and validated in Korea.
|
2. Materials and Methods 2.4. Diagnosis of NAFLD We used the NAFLD-liver fat score to diagnose NAFLD [13], a tool that has undergone validation in the Korean population [14]. Reference 14. Lee, Y.H.; Bang, H.; Park, Y.M.; Bae, J.C.; Lee, B.W.; Kang, E.S.; Cha, B.S.; Lee, H.C.; Balkau, B.; Lee, W.Y.; et al. Non-laboratory-based self-assessment screening score for non-alcoholic fatty liver disease: development, validation and comparison with other scores. PLoS One 2014, 9, e107584, doi:10.1371/journal.pone.0107584. |
2)Another issue is that NAFLD-liver fat score is unable to tell the severity of liver fat or fibrosis deposition in the liver. It is unable to tell whether the incident NAFLD is a mild, moderate or severe fatty liver or NASH. The clinical significance of NAFLD was not as much as incident Diabetes, which was used to calculate the NAFLD-liver fat score. If the authors were unable to find imaging data to define NAFLD, then a proper discussion of limitations and a mild tone in the conclusion should be used.
Response: Thank you for your valuable comment. You have raised an important issue regarding the limitations of the NAFLD-liver fat score in assessing the severity of liver fat or fibrosis deposition. Unfortunately, imaging data to define NAFLD was not available in the KoGES, we add the limitation and revised the conclusion as follows;
|
Line 364 Second, we defined NAFLD using a surrogate marker (NAFLD-liver fat score) rather than imaging modalities (abdominal ultrasonography or magnetic resonance image). The NAFLD-liver fat score lacks the ability to differentiate between mild, moderate, or severe fatty liver or NASH in incident NAFLD. Therefore, this study did not assess the severity of liver fat or fibrosis deposition, which limits our understanding of the clinical implications and disease progression within the NAFLD spectrum.
Line 378 In total population, persistent AO and progression to AO are associated with higher risk for NAFLD. Persistent AO was a significant risk factor for developing NAFLD only in women, suggesting that both maintaining lean WC or improvement from AO would be effective strategies for preventing NAFLD in women while maintaining lean WC is more crucial in men. A health strategy that focuses on maintaining a healthy WC throughout life through physical activity and a healthy diet is likely to be more effective in preventing NAFLD than solely relying on reducing WC at a later stage. Also, considering gender-specific risk profiles can ultimately contribute to the development of more effective health policies and strategies for addressing NAFLD and related health concerns. Additional research is warranted to comprehensively assess the severity of NAFLD, in order to obtain a more precise understanding of the relationship between AO and the risk of NAFLD. |
the authors have excluded more than 2000 participants with previous NAFLD, are they defined by ultrasound or NAFLD-liver fat score? Please specify. The authors has excluded more than half of the total cohort (10,030), sensitivity analysis should be expended.
Response: The participants who were excluded due to previous NAFLD were also identified using the NAFLD-liver fat score. In response to the valuable suggestion from reviewer #2, we conducted a sensitivity analysis for incident metabolic dysfunction-associated fatty liver disease (MAFLD) among 5360 participants. In this analysis, we did not exclude participants with viral hepatitis infection or heavy alcohol consumption, as indicated in Table for reviewers 1. The risk of incident MAFLD remained consistent with the original analysis.
Table for reviewers 1. Association of AO transition status and the risk of incident metabolic dysfunction-associated fatty liver disease among 5360 participants.
|
  |
|
Unadjusted |
Model 1 |
Model 2 |
Model 3 |
||||
|
  |
|
HR (95% CI) |
p-value |
HR (95% CI) |
p-value |
HR (95% CI) |
p-value |
HR (95% CI) |
p-value |
|
Overall |
Persistent lean |
ref |
|
ref |
|
ref |
|
ref |
|
|
Improved from abdominal obesity |
1.409(1.168-1.700) |
<.001 |
1.065(0.878-1.292) |
0.520 |
1.060(0.866-1.297) |
0.572 |
1.058(0.864-1.296) |
0.585 |
|
|
Progressed to abdominal obesity |
2.235(1.963-2.545) |
<.001 |
1.771(1.548-2.024) |
<.001 |
1.790(1.555-2.061) |
<.001 |
1.685(1.461-1.942) |
<.001 |
|
|
  |
Persistent abdominal obesity |
2.698(2.428-2.998) |
<.001 |
1.413(1.232-1.621) |
<.001 |
1.364(1.178-1.578) |
<.001 |
1.226(1.057-1.422) |
0.007 |
|
Men |
Persistent lean |
ref |
|
ref |
|
ref |
|
ref |
|
|
|
Improved from abdominal obesity |
1.978(1.456-2.688) |
<.001 |
1.218(0.889-1.669) |
0.221 |
1.214(0.882-1.672) |
0.234 |
1.334(0.968-1.838) |
0.079 |
|
|
Progressed to abdominal obesity |
2.168(1.762-2.669) |
<.001 |
1.514(1.222-1.876) |
<.001 |
1.518(1.213-1.900) |
<.001 |
1.411(1.122-1.773) |
0.003 |
|
|
Persistent abdominal obesity |
2.788(2.289-3.397) |
<.001 |
1.150(0.906-1.461) |
0.250 |
1.064(0.829-1.365) |
0.625 |
1.003(0.779-1.291) |
0.981 |
|
Women |
Persistent lean |
ref |
|
ref |
|
ref |
|
ref |
|
|
|
Improved from abdominal obesity |
1.367(1.075-1.737) |
0.011 |
1.009(0.791-1.288) |
0.941 |
0.989(0.763-1.282) |
0.935 |
0.949(0.730-1.232) |
0.692 |
|
|
Progressed to abdominal obesity |
2.479(2.092-2.936) |
<.001 |
1.927(1.619-2.293) |
<.001 |
1.939(1.613-2.331) |
<.001 |
1.861(1.547-2.240) |
<.001 |
|
|
Persistent abdominal obesity |
3.018(2.644-3.446) |
<.001 |
1.506(1.268-1.789) |
<.001 |
1.503(1.248-1.810) |
<.001 |
1.367(1.133-1.650) |
0.001 |
Model 1: adjusted for age, sex (in total population), and BMI; model 2: model 1 plus smoking, physical activity, alcohol consumption, total energy intake; model 3: model 2 plus mean blood pressure, glycosylated hemoglobin, total cholesterol, high-sensitivity C-reactive protein, and alanine aminotransferase.
We also conducted an additional sensitivity analysis for incident NAFLD, using a hepatic steatosis index >36, as well as employing another exclusion criterion that conforms to NAFLD among 3858 participants, as shown in Table for reviewers 2. We found that the results were consistent with the original findings.
Table for reviewer 2. Association of AO transition status and the risk of incident NAFLD defined as HSI >36 among 3858 participants.
|
  |
|
Unadjusted |
Model 1 |
Model 2 |
Model 3 |
||||
|
  |
|
HR (95% CI) |
p-value |
HR (95% CI) |
p-value |
HR (95% CI) |
p-value |
HR (95% CI) |
p-value |
|
Overall |
Persistent lean |
ref |
|
ref |
|
ref |
|
ref |
|
|
Improved from abdominal obesity |
1.481(1.198-1.830) |
<.001 |
1.430(1.157-1.768) |
0.001 |
1.452(1.160-1.817) |
0.001 |
1.508(1.201-1.892) |
<.001 |
|
|
Progressed to abdominal obesity |
1.630(1.375-1.932) |
<.001 |
1.558(1.314-1.847) |
<.001 |
1.562(1.307-1.868) |
<.001 |
1.562(1.301-1.876) |
<.001 |
|
|
  |
Persistent abdominal obesity |
2.324(2.032-2.657) |
<.001 |
2.159(1.886-2.472) |
<.001 |
2.119(1.837-2.445) |
<.001 |
2.093(1.802-2.431) |
<.001 |
|
Men |
Persistent lean |
ref |
|
ref |
|
ref |
|
ref |
|
|
|
Improved from abdominal obesity |
1.749(1.340-2.281) |
<.001 |
1.831(1.403-2.390) |
<.001 |
1.915(1.452-2.525) |
<.001 |
1.948(1.473-2.577) |
<.001 |
|
|
Progressed to abdominal obesity |
1.807(1.463-2.231) |
<.001 |
1.809(1.465-2.234) |
<.001 |
1.815(1.460-2.255) |
<.001 |
1.835(1.471-2.290) |
<.001 |
|
|
Persistent abdominal obesity |
2.652(2.245-3.132) |
<.001 |
2.659(2.251-3.140) |
<.001 |
2.599(2.182-3.097) |
<.001 |
2.548(2.121-3.060) |
<.001 |
|
Women |
Persistent lean |
ref |
|
ref |
|
ref |
|
ref |
|
|
|
Improved from abdominal obesity |
1.034(0.729-1.467) |
0.852 |
1.033(0.728-1.465) |
0.858 |
1.019(0.693-1.499) |
0.925 |
1.079(0.727-1.602) |
0.705 |
|
|
Progressed to abdominal obesity |
1.328(0.996-1.771) |
0.053 |
1.267(0.947-1.695) |
0.112 |
1.262(0.918-1.735) |
0.152 |
1.298(0.932-1.809) |
0.123 |
|
|
Persistent abdominal obesity |
1.770(1.410-2.223) |
<.001 |
1.645(1.295-2.089) |
<.001 |
1.622(1.253-2.100) |
<.001 |
1.679(1.280-2.201) |
<.001 |
Model 1: adjusted for age, sex (in total population), and BMI; model 2: model 1 plus smoking, physical activity, alcohol consumption, total energy intake; model 3: model 2 plus mean blood pressure, glycosylated hemoglobin, total cholesterol, high-sensitivity C-reactive protein, and alanine aminotransferase.
Minor issues are:
1)Please present the comparison of clinical characteristics among the four groups (Persistent lean, Improved from abdominal obesity, Progressed to abdominal obesity, Persistent abdominal obesity).
Response: In accordance with reviewer #2’s suggestion, we presented the comparison of clinical characteristics among the four groups in total population, men, and women, respectively, as shown in Table for reviewers 3.
Table for reviewers 3. Clinical characteristics of the study population based the transition status of AO.
1) Overall
|
|
Characteristics |
Total (n=4467) |
Persistent lean WC (n=3363) (1) |
Improved AO (n=198) (2) |
Progressed to AO (n=376) (3) |
Persistent AO (n=530) (4) |
p-value |
(1) vs (2) |
(1) vs (3) |
(1) vs (4) |
(2) vs (3) |
(2) vs (4) |
(3) vs (4) |
|
Overall |
Age, years |
51.555±8.750 |
50.652±8.558 |
51.222±8.742 |
53.551±8.849 |
55.996±8.314 |
<.0001 |
0.3626 |
<.0001 |
<.0001 |
0.002 |
<.0001 |
<.0001 |
|
|
Waist circumference, cm |
79.596±7.871 |
76.733±6.098 |
90.040±3.541 |
82.626±3.758 |
91.709±5.048 |
<.0001 |
<.0001 |
<.0001 |
<.0001 |
<.0001 |
0.0005 |
<.0001 |
|
|
Body fat, % |
26.039±7.145 |
24.409±6.716 |
30.837±5.431 |
28.417±6.292 |
32.925±5.322 |
<.0001 |
<.0001 |
<.0001 |
<.0001 |
<.0001 |
0.0001 |
<.0001 |
|
|
BMI, kg/m2 |
23.707±2.789 |
22.973±2.402 |
25.295±2.424 |
24.701±2.103 |
27.068±2.702 |
<.0001 |
<.0001 |
<.0001 |
<.0001 |
0.0053 |
<.0001 |
<.0001 |
|
|
Skeletal muscle mass, kg |
41.927±7.483 |
42.050±7.448 |
41.113±7.208 |
41.462±7.900 |
41.782±7.490 |
0.1805 |
0.0876 |
0.1489 |
0.4442 |
0.5961 |
0.2848 |
0.5274 |
|
|
Body weight, kg |
60.133±8.934 |
58.930±8.477 |
62.894±8.718 |
61.202±8.319 |
65.984±9.621 |
<.0001 |
<.0001 |
<.0001 |
<.0001 |
0.0256 |
<.0001 |
<.0001 |
|
|
Smoking status |
|
|
|
|
|
<.0001 |
<.0001 |
<.0001 |
<.0001 |
0.0066 |
0.0794 |
0.0035 |
|
|
non |
2978(67.47) |
2099(63.01) |
163(84.02) |
276(75.41) |
440(84.13) |
|
|
|
|
|
|
|
|
|
ex |
562(12.73) |
484(14.53) |
18(9.28) |
33(9.02) |
27(5.16) |
|
|
|
|
|
|
|
|
|
intermittent |
81(1.84) |
61(1.83) |
4(2.06) |
5(1.37) |
11(2.10) |
|
|
|
|
|
|
|
|
|
daily |
793(17.97) |
687(20.62) |
9(4.64) |
52(14.21) |
45(8.60) |
|
|
|
|
|
|
|
|
|
Physical activity |
|
|
|
|
|
<.0001 |
0.0002 |
<.0001 |
<.0001 |
<.0001 |
<.0001 |
0.0506 |
|
|
low |
306(7.12) |
219(6.75) |
22(11.46) |
36(10.20) |
29(5.74) |
|
|
|
|
|
|
|
|
|
mod |
2601(60.54) |
2071(63.80) |
137(71.35) |
159(45.04) |
234(46.34) |
|
|
|
|
|
|
|
|
|
high |
1389(32.33) |
956(29.45) |
33(17.19) |
158(44.76) |
242(47.92) |
|
|
|
|
|
|
|
|
|
Alcohol consumption |
|
|
|
|
|
<.0001 |
0.0414 |
0.0032 |
<.0001 |
0.9055 |
0.1299 |
0.0872 |
|
|
No |
2524(56.96) |
1815(54.36) |
120(61.86) |
232(62.37) |
357(67.87) |
|
|
|
|
|
|
|
|
|
Yes |
1907(43.04) |
1524(45.64) |
74(38.14) |
140(37.63) |
169(32.13) |
|
|
|
|
|
|
|
|
|
Total energy intake, kcal/day |
1934.140±705.464 |
1926.836±666.265 |
1879.534±767.781 |
1994.334±929.209 |
1957.851±737.737 |
0.1929 |
0.3713 |
0.0818 |
0.3574 |
0.0695 |
0.1937 |
0.4503 |
|
|
Mean blood pressure, mmHg |
93.568±12.429 |
92.532±12.220 |
92.987±11.914 |
96.199±13.108 |
98.488±12.013 |
<.0001 |
0.6124 |
<.0001 |
<.0001 |
0.0029 |
<.0001 |
0.0056 |
|
|
Fasting glucose, mg/dl |
82.469±12.171 |
82.468±12.393 |
81.566±10.006 |
81.330±10.053 |
83.621±12.764 |
0.0275 |
0.3105 |
0.0854 |
0.0426 |
0.8252 |
0.0426 |
0.0052 |
|
|
Total cholesterol, mg/dl |
187.504±33.256 |
186.517±33.014 |
189.470±35.448 |
186.790±33.830 |
193.536±32.957 |
<.0001 |
0.2239 |
0.88 |
<.0001 |
0.3578 |
0.1414 |
0.0026 |
|
|
hsCRP |
0.226±0.629 |
0.210±0.489 |
0.199±0.256 |
0.281±0.647 |
0.303±1.220 |
0.0037 |
0.8147 |
0.037 |
0.0014 |
0.1371 |
0.0462 |
0.5985 |
|
|
ALT |
21.889±8.698 |
22.200±9.123 |
21.384±7.706 |
21.402±7.915 |
20.445±6.339 |
0.0001 |
0.1985 |
0.0908 |
<.0001 |
0.9814 |
0.1943 |
0.1023 |
|
|
AST |
26.343±6.774 |
26.577±7.022 |
25.424±4.767 |
25.926±6.331 |
25.494±5.969 |
0.0006 |
0.0198 |
0.0766 |
0.0006 |
0.3986 |
0.901 |
0.3444 |
|
|
HOMA-IR |
1.320±0.595 |
1.294±0.586 |
1.323±0.587 |
1.401±0.579 |
1.421±0.646 |
<.001 |
0.5111 |
0.001 |
<.001 |
0.1359 |
0.0463 |
0.6028 |
|
|
HOMA-beta |
151.392±136.726 |
150.020±142.570 |
160.054±113.915 |
162.690±113.323 |
148.877±120.578 |
0.2761 |
0.3168 |
0.0888 |
0.8581 |
0.8266 |
0.3274 |
0.1345 |
|
|
Insulin |
6.479±2.772 |
6.351±2.723 |
6.603±2.876 |
6.975±2.787 |
6.894±2.948 |
<.001 |
0.2132 |
<.001 |
<.001 |
0.1253 |
0.2068 |
0.6622 |
2) Men
|
|
Characteristics |
Total (n=1867) |
Persistent lean WC (n=1607) (1) |
Improved AO (n=50) (2) |
Progressed to AO (n=115) (3) |
Persistent AO (n=95) (4) |
p-value |
(1) vs (2) |
(1) vs (3) |
(1) vs (4) |
(2) vs (3) |
(2) vs (4) |
(3) vs (4) |
|
Men |
Age, years |
52.147±8.779 |
51.866±8.767 |
53.400±8.519 |
54.070±8.946 |
53.916±8.495 |
0.0072 |
0.2228 |
0.0092 |
0.0268 |
0.6518 |
0.7361 |
0.8992 |
|
|
Waist circumference, cm |
80.800±6.739 |
79.303±5.757 |
91.960±2.241 |
85.997±2.801 |
93.971±3.222 |
<.0001 |
<.0001 |
<.0001 |
<.0001 |
<.0001 |
0.0349 |
<.0001 |
|
|
Body fat, % |
20.130±4.880 |
19.503±4.583 |
24.896±4.998 |
21.874±4.597 |
26.179±4.008 |
<.0001 |
<.0001 |
<.0001 |
<.0001 |
0.0001 |
0.1112 |
<.0001 |
|
|
BMI, kg/m2 |
23.331±2.571 |
22.878±2.326 |
25.927±2.123 |
25.018±1.739 |
27.602±1.956 |
<.0001 |
<.0001 |
<.0001 |
<.0001 |
0.0191 |
<.0001 |
<.0001 |
|
|
Skeletal muscle mass, kg |
48.589±5.824 |
47.973±5.634 |
51.235±5.352 |
51.391±4.936 |
54.253±5.865 |
<.0001 |
<.0001 |
<.0001 |
<.0001 |
0.8698 |
0.0022 |
0.0002 |
|
|
Body weight, kg |
64.613±8.574 |
63.250±7.927 |
72.267±6.540 |
69.657±5.770 |
77.624±7.639 |
<.0001 |
<.0001 |
<.0001 |
<.0001 |
0.0489 |
<.0001 |
<.0001 |
|
|
Smoking status |
|
|
|
|
|
0.0032 |
0.0011 |
0.8051 |
0.0384 |
0.0026 |
0.0171 |
0.4516 |
|
|
non |
501(26.96) |
424(26.50) |
22(44.00) |
27(23.89) |
28(29.47) |
|
|
|
|
|
|
|
|
|
ex |
545(29.33) |
472(29.50) |
18(36.00) |
32(28.32) |
23(24.21) |
|
|
|
|
|
|
|
|
|
intermittent |
67(3.61) |
51(3.19) |
3(6.00) |
5(4.42) |
8(8.42) |
|
|
|
|
|
|
|
|
|
daily |
745(40.10) |
653(40.81) |
7(14.00) |
49(43.36) |
36(37.89) |
|
|
|
|
|
|
|
|
|
Physical activity |
|
|
|
|
|
<.0001 |
0.0157 |
<.0001 |
0.0045 |
<.0001 |
<.0001 |
0.1494 |
|
|
low |
98(5.47) |
82(5.31) |
3(6.12) |
10(9.52) |
3(3.26) |
|
|
|
|
|
|
|
|
|
mod |
1017(56.82) |
902(58.42) |
38(77.55) |
37(35.24) |
40(43.48) |
|
|
|
|
|
|
|
|
|
high |
675(37.71) |
560(36.27) |
8(16.33) |
58(55.24) |
49(53.26) |
|
|
|
|
|
|
|
|
|
Alcohol consumption |
|
|
|
|
|
0.4018 |
0.1706 |
0.8994 |
0.3357 |
0.2829 |
0.0875 |
0.4068 |
|
|
No |
657(35.42) |
565(35.38) |
22(44.90) |
41(35.96) |
29(30.53) |
|
|
|
|
|
|
|
|
|
Yes |
1198(64.58) |
1032(64.62) |
27(55.10) |
73(64.04) |
66(69.47) |
|
|
|
|
|
|
|
|
|
Total energy intake, kcal/day |
1991.574±671.299 |
1977.268±646.092 |
1973.773±669.903 |
2098.156±923.697 |
2114.745±714.901 |
0.0846 |
0.9714 |
0.0656 |
0.0575 |
0.279 |
0.2356 |
0.8609 |
|
|
Mean blood pressure, mmHg |
95.117±11.716 |
94.713±11.652 |
96.573±10.399 |
96.655±12.855 |
99.319±11.114 |
0.0007 |
0.2672 |
0.085 |
0.0002 |
0.967 |
0.1783 |
0.0999 |
|
|
Fasting glucose, mg/dl |
84.253±13.259 |
84.268±13.575 |
85.900±13.796 |
82.270±10.485 |
85.537±10.025 |
0.237 |
0.3912 |
0.1185 |
0.3645 |
0.1061 |
0.8754 |
0.0756 |
|
|
Total cholesterol, mg/dl |
189.197±33.332 |
189.073±33.291 |
200.360±33.559 |
184.043±35.212 |
191.663±30.399 |
0.0305 |
0.0183 |
0.1176 |
0.4611 |
0.0038 |
0.1349 |
0.0988 |
|
|
hsCRP |
0.235±0.520 |
0.226±0.488 |
0.255±0.317 |
0.342±0.970 |
0.236±0.317 |
0.1484 |
0.7012 |
0.0218 |
0.8663 |
0.3259 |
0.8308 |
0.1417 |
|
|
ALT |
25.567±9.843 |
25.515±9.997 |
26.000±8.074 |
26.609±9.710 |
24.947±8.101 |
0.614 |
0.7318 |
0.2501 |
0.585 |
0.7152 |
0.5407 |
0.2238 |
|
|
AST |
27.988±7.242 |
28.049±7.360 |
26.640±5.236 |
28.174±6.596 |
27.442±6.871 |
0.4824 |
0.1756 |
0.8584 |
0.4274 |
0.2114 |
0.5263 |
0.4662 |
|
|
HOMA-IR |
1.237±0.567 |
1.228±0.560 |
1.272±0.606 |
1.336±0.529 |
1.250±0.701 |
0.2459 |
0.594 |
0.0485 |
0.7109 |
0.5009 |
0.8304 |
0.2748 |
|
|
HOMA-beta |
127.824±125.483 |
128.427±130.460 |
111.803±71.955 |
149.653±94.665 |
99.692±80.746 |
0.0284 |
0.3603 |
0.0806 |
0.03 |
0.0771 |
0.5826 |
0.0041 |
|
|
Insulin |
5.944±2.588 |
5.895±2.536 |
5.990±2.756 |
6.618±2.661 |
5.936±3.154 |
0.0384 |
0.7983 |
0.0038 |
0.8816 |
0.1515 |
0.9045 |
0.057 |
3) Women
|
|
Characteristics |
Total (n=2600) |
Persistent lean WC (n=1756) (1) |
Improved AO (n=148) (2) |
Progressed to AO (n=261) (3) |
Persistent AO (n=435) (4) |
p-value |
(1) vs (2) |
(1) vs (3) |
(1) vs (4) |
(2) vs (3) |
(2) vs (4) |
(3) vs (4) |
|
Women |
Age, years |
51.130±8.706 |
49.541±8.210 |
50.486±8.721 |
53.322±8.813 |
56.451±8.214 |
<.0001 |
0.1835 |
<.0001 |
<.0001 |
0.0009 |
<.0001 |
<.0001 |
|
|
Waist circumference, cm |
78.731±8.488 |
74.382±5.416 |
89.391±3.668 |
81.141±3.113 |
91.216±5.239 |
<.0001 |
<.0001 |
<.0001 |
<.0001 |
<.0001 |
0.0002 |
<.0001 |
|
|
Body fat, % |
30.275±5.253 |
28.891±5.004 |
32.804±3.939 |
31.310±4.523 |
34.393±4.352 |
<.0001 |
<.0001 |
<.0001 |
<.0001 |
0.0025 |
0.0005 |
<.0001 |
|
|
BMI, kg/m2 |
23.977±2.906 |
23.060±2.467 |
25.086±2.487 |
24.562±2.234 |
26.951±2.828 |
<.0001 |
<.0001 |
<.0001 |
<.0001 |
0.0426 |
<.0001 |
<.0001 |
|
|
Skeletal muscle mass, kg |
37.152±4.149 |
36.639±3.977 |
37.762±3.804 |
37.070±4.032 |
39.068±4.428 |
<.0001 |
0.0012 |
0.1095 |
<.0001 |
0.0974 |
0.0007 |
<.0001 |
|
|
Body weight, kg |
56.918±7.721 |
54.976±6.888 |
59.791±6.956 |
57.476±6.297 |
63.442±8.003 |
<.0001 |
<.0001 |
<.0001 |
<.0001 |
0.0014 |
<.0001 |
<.0001 |
|
|
Smoking status |
|
|
|
|
|
0.8766 |
0.8122 |
0.7277 |
0.8421 |
0.7401 |
0.8378 |
0.4748 |
|
|
non |
2477(96.91) |
1675(96.76) |
141(97.92) |
249(98.42) |
412(96.26) |
|
|
|
|
|
|
|
|
|
ex |
17(0.67) |
12(0.69) |
0(0.00) |
1(0.40) |
4(0.93) |
|
|
|
|
|
|
|
|
|
intermittent |
14(0.55) |
10(0.58) |
1(0.69) |
0(0.00) |
3(0.70) |
|
|
|
|
|
|
|
|
|
daily |
48(1.88) |
34(1.96) |
2(1.39) |
3(1.19) |
9(2.10) |
|
|
|
|
|
|
|
|
|
Physical activity |
|
|
|
|
|
<.0001 |
0.0446 |
<.0001 |
<.0001 |
<.0001 |
<.0001 |
0.08 |
|
|
low |
208(8.30) |
137(8.05) |
19(13.29) |
26(10.48) |
26(6.30) |
|
|
|
|
|
|
|
|
|
mod |
1584(63.21) |
1169(68.68) |
99(69.23) |
122(49.19) |
194(46.97) |
|
|
|
|
|
|
|
|
|
high |
714(28.49) |
396(23.27) |
25(17.48) |
100(40.32) |
193(46.73) |
|
|
|
|
|
|
|
|
|
Alcohol consumption |
|
|
|
|
|
0.1483 |
0.2855 |
0.4474 |
0.0701 |
0.168 |
0.0433 |
0.5417 |
|
|
No |
1867(72.48) |
1250(71.76) |
98(67.59) |
191(74.03) |
328(76.10) |
|
|
|
|
|
|
|
|
|
Yes |
709(27.52) |
492(28.24) |
47(32.41) |
67(25.97) |
103(23.90) |
|
|
|
|
|
|
|
|
|
Total energy intake, kcal/day |
1892.960±726.317 |
1880.997±681.039 |
1846.313±799.024 |
1948.911±929.760 |
1923.448±739.043 |
0.3494 |
0.5882 |
0.163 |
0.2855 |
0.1801 |
0.2786 |
0.6591 |
|
|
Mean blood pressure, mmHg |
92.455±12.803 |
90.536±12.388 |
91.775±12.178 |
95.997±13.237 |
98.307±12.205 |
<.0001 |
0.2447 |
<.0001 |
<.0001 |
0.001 |
<.0001 |
0.0178 |
|
|
Fasting glucose, mg/dl |
81.188±11.153 |
80.821±10.950 |
80.101±7.886 |
80.916±9.848 |
83.202±13.260 |
0.0005 |
0.45 |
0.8975 |
<.0001 |
0.4768 |
0.0034 |
0.0087 |
|
|
Total cholesterol, mg/dl |
186.288±33.154 |
184.179±32.593 |
185.791±35.420 |
188.000±33.199 |
193.945±33.509 |
<.0001 |
0.568 |
0.0808 |
<.0001 |
0.515 |
0.0094 |
0.0214 |
|
|
hsCRP |
0.221±0.696 |
0.195±0.489 |
0.180±0.229 |
0.254±0.436 |
0.318±1.338 |
0.007 |
0.8068 |
0.1956 |
0.0009 |
0.2994 |
0.037 |
0.2413 |
|
|
ALT |
19.247±6.613 |
19.166±6.980 |
19.824±6.940 |
19.107±5.634 |
19.462±5.419 |
0.5797 |
0.2452 |
0.893 |
0.4038 |
0.2922 |
0.565 |
0.4934 |
|
|
AST |
25.161±6.153 |
25.229±6.410 |
25.014±4.543 |
24.935±5.959 |
25.069±5.674 |
0.8609 |
0.6819 |
0.4706 |
0.6263 |
0.9012 |
0.9246 |
0.7808 |
|
|
HOMA-IR |
1.379±0.607 |
1.355±0.604 |
1.340±0.582 |
1.429±0.598 |
1.459±0.628 |
0.0055 |
0.7764 |
0.066 |
0.0014 |
0.1552 |
0.04 |
0.5297 |
|
|
HOMA-beta |
168.303±141.886 |
169.780±150.153 |
176.029±120.742 |
168.384±120.296 |
159.643±125.166 |
0.5232 |
0.607 |
0.8821 |
0.1828 |
0.6006 |
0.2252 |
0.4317 |
|
|
Insulin |
6.864±2.836 |
6.769±2.821 |
6.810±2.896 |
7.132±2.832 |
7.103±2.863 |
0.059 |
0.8654 |
0.0534 |
0.0279 |
0.2694 |
0.2779 |
0.8945 |
please compare HOMA-IR and HOMA-B, fasting glucose, and Insulin in all tables.
Response: As fasting glucose was already shown in Table 1 and Table 2, we added the results for the comparison of HOMA-IR and HOMA-B, and Insulin in the revised Table 1 and Table 2.
Table 1. Baseline characteristics of the study population.
|
Variables |
Total (n=4,467) |
Men (n=1,867) |
Women (n=2,600) |
P-value |
|
Fasting glucose, mg/dL |
82.5 ± 12.2 |
84.3 ± 13.3 |
81.2 ± 11.2 |
<0.001 |
|
HOMA-IR |
1.320 ± 0.595 |
1.237 ± 0.567 |
1.379 ± 0.607 |
<0.001 |
|
HOMA-beta |
151.392 ± 136.726 |
127.824 ± 125.483 |
168.303 ± 141.886 |
<0.001 |
|
Insulin |
6.479 ± 2.772 |
5.944 ± 2.588 |
6.864 ± 2.836 |
<0.001 |
Abbreviations: ALT, alanine aminotransferase; AST, aspartate aminotransferase; BMI, body mass index; hsCRP, high-sensitivity C-reactive protein; WC, waist circumference; HOMA-IR, homeostatic model assessment for insulin resistance.
Table 2. Baseline characteristics of men and women according to incident non-alcoholic fatty liver disease.
|
|
Total |
|
|
Men |
|
|
Women |
|
|
|
|
No incident NAFLD (n=2642) |
Incident NAFLD (n=1825) |
P-value |
No incident NAFLD (n=1,110) |
Incident NAFLD (n=757) |
P-value |
No incident NAFLD (n=1,532) |
Incident NAFLD (n=1,068) |
P-value |
|
Fasting glucose, mg/dL |
80.9 ± 8.4 |
84.7 ± 15.9 |
<.001 |
82.4 ± 9.4 |
87.0 ± 17.1 |
<.001 |
79.9 ± 7.4 |
83.0 ± 14.8 |
<.001 |
|
HOMA-IR |
1.261 ± 0.561 |
1.405 ± 0.631 |
<0.001 |
1.177 ± 0.533 |
1.326 ± 0.603 |
<0.001 |
1.321 ± 0.572 |
1.462 ± 0.645 |
<.0001 |
|
HOMA-beta |
155.186 ± 137.717 |
145.890 ± 135.126 |
0.026 |
133.450 ± 144.596 |
119.553 ± 89.774 |
0.011 |
170.934 ± 130.311 |
164.525 ± 157.008 |
0.273 |
|
Insulin |
6.288 ± 2.677 |
6.757 ± 2.883 |
<0.001 |
5.759 ± 2.496 |
6.217 ± 2.697 |
<0.001 |
6.671 ± 2.738 |
7.140 ± 2.949 |
<.0001 |
Abbreviations: ALT, alanine aminotransferase; AST, aspartate aminotransferase; BMI, body mass index; hsCRP, high-sensitivity C-reactive protein; WC, waist circumference; HOMA-IR, homeostatic model assessment for insulin resistance.
current table 3: Please change covariate FPG to HbA1c if it is possible. Try to adjust BMI if possible.
Response: In accordance with reviewer #2’s suggestion, we changed covariate FPG to HbA1c and adjusted BMI as a covariate in the revised Table 3 as below:
Table 3. Cox proportional hazards regression analysis for incident non-alcoholic fatty liver disease according to abdominal obesity patterns.
|
  |
|
Unadjusted |
Model 1 |
Model 2 |
Model 3 |
||||
|
  |
|
HR (95% CI) |
p-value |
HR (95% CI) |
p-value |
HR (95% CI) |
p-value |
HR (95% CI) |
p-value |
|
Total |
Persistent lean WC |
ref |
|
ref |
|
ref |
|
ref |
|
|
Improved from abdominal obesity |
1.39 (1.13–1.72) |
0.002 |
1.03 (0.83–1.27) |
0.822 |
1.02 (0.81–1.28) |
0.852 |
1.06 (0.84–1.33) |
0.637 |
|
|
Progressed to abdominal obesity |
2.26 (1.97–2.61) |
<.001 |
1.73 (1.52–2.03) |
<.001 |
1.77 (1.52–2.06) |
<.001 |
1.73 (1.48–2.02) |
<.001 |
|
|
  |
Persistent abdominal obesity |
2.56 (2.26–2.89) |
<.001 |
1.32 (1.13–1.54) |
<.001 |
1.29 (1.09–1.52) |
0.003 |
1.33 (1.13–1.57) |
<.001 |
|
Men |
Persistent lean WC |
ref |
|
ref |
|
ref |
|
ref |
|
|
|
Improved from abdominal obesity |
1.91 (1.32–2.77) |
0.001 |
1.30 (0.89–1.90) |
0.180 |
1.34 (0.90–1.98) |
0.146 |
1.47 (0.99–2.18) |
0.055 |
|
|
Progressed to abdominal obesity |
2.25 (1.77–2.86) |
<.001 |
1.60 (1.25–2.05) |
<.001 |
1.57 (1.21–2.05) |
<.001 |
1.60 (1.22–2.09) |
<.001 |
|
|
Persistent abdominal obesity |
2.78 (2.14–3.61) |
<.001 |
1.20 (0.88–1.64) |
0.249 |
1.12 (0.81–1.54) |
0.505 |
1.21 (0.87–1.69) |
0.253 |
|
Women |
Persistent lean WC |
ref |
|
ref |
|
ref |
|
ref |
|
|
|
Improved from abdominal obesity |
1.33 (1.03–1.73) |
0.029 |
0.95 (0.73–1. 23) |
0.691 |
0.93 (0.70–1.23) |
0.586 |
0.93 (0.71–1.24) |
0.628 |
|
|
Progressed to abdominal obesity |
2.38 (2.00–2.84) |
<.001 |
1.81 (1.51–2.17) |
<.001 |
1.83 (1.51–2.21) |
<.001 |
1.78 (1.47–2.16) |
<.001 |
|
|
Persistent abdominal obesity |
2.70 (2.33–3.12) |
<.001 |
1.31 (1.09–1.58) |
0.004 |
1.31 (1.07–1.59) |
0.008 |
1.36 (1.12–1.65) |
0.002 |
Model 1: adjusted for age, sex (in total population), and body mass index; model 2: model 1 plus smoking, physical activity, alcohol consumption, total energy intake; model 3: model 2 plus mean blood pressure, glycosylated hemoglobin, total cholesterol, high-sensitivity C-reactive protein, and alanine aminotransferase.
|
2. Materials and Methods 2.5. Statistical analysis In model 1, we adjusted for sex, age, and BMI. In model 2, we adjusted for variables used in model 1 plus smoking status, alcohol consumption status, physical activity, and total energy intake. In model 3, we adjusted for variables used in model 2 plus MBP, HbA1c, serum total cholesterol, CRP, and ALT levels.
3. Results Table 3 presents the HR (95% CI) for NAFLD incidence according to the longitudinal AO patterns using Cox proportional hazards regression model. Compared with the reference persistent lean WC group, the HRs (95% CI) for NAFLD incidence in improved AO, progressed to AO, and persistent AO groups were 1.39 (1.13–1.72), 2.26 (1.97–2.61), and 2.56 (2.26–2.89), respectively. In model 3, compared to persistent lean WC group, the fully adjusted HRs (95% CI) for NAFLD incidence in improved AO, progressed to AO, and persistent AO groups were 1.06 (0.84–1.33), 1.73 (1.48–2.02), and 1.33 (1.13–1.57), respectively. In pairwise comparison analysis, both progressed to AO and persistent AO groups had significantly higher risk for developing NAFLD than improved AO group. There was no difference in the risk for developing NAFLD between progressed to AO and persistent AO groups (Supplementary Table 2). In men, compared to the reference persistent lean WC group, the HRs (95% CI) for NAFLD incidence in improved AO, progressed to AO, and persistent AO groups were 1.91 (1.32–2.77), 2.25 (1.77–2.86), and 2.78 (2.14–3.61), respectively. In progress to AO group, these significant associations were consistently noticed in models 1, 2, and 3. In pairwise comparison analysis, progressed to AO group had a significantly higher risk for developing NAFLD than persistent lean WC group. There was no difference in the risk for developing NAFLD between progressed to AO and persistent AO groups and between improved AO and persistent AO groups (Supplementary Table 2).In women, compared to the reference persistent lean WC group, the HRs (95% CI) for NAFLD incidence in improved AO, progressed to AO, and persistent AO groups were 1.33 (1.03–1.73), 2.38 (2.00–2.84), and 2.70 (2.33–3.12), respectively. In progress to AO and persistent AO groups, these associations remained statistically significant in adjusted models. In pairwise comparison analysis, women showed similar patterns of association as those in the total population (Supplementary Table 2).
4. Discussion In this study, we found that participants who had persistent AO, progressed to AO, or improved AO had significantly higher risk for NAFLD incidence compared to persistently lean WC participants. Men who had progressed to AO over two years had significantly higher risk for NAFLD than those without any AO. Women who had persistent AO or progressed to AO had significantly higher risk for NAFLD than those who had no AO or improved AO. These associations were noticed even when the observation period was extended for four years. |
I did not see Figure 1 in the manuscript.
Response: We really apologize the omitting figure 1. We added the Figure 1.
Figure 1. Flow chart of the study population.
The name of the four groups is a bit misleading, persistent lean indicates normal BMI rather than WC.
Response: Thank you for your valuable comment. We have revised ‘lean’ to ‘lean WC’ throughout the manuscript.

Reviewer 3 Report
The manuscript is well written. All sections appropriately provide detailed information, except conclusion (see further comments). Methods and results are well described, tehnical editing is required (explained in further comments). Researchers provided an interesting insight into correlation of NAFLD and abdominal obesity.
In abstract (an throught the study) consider a better term for a "improved from AO", perhaps just "improved AO"
LINE 42 Decode this as interleukin 6 (IL-6) and tumor necrosis factor α (TNF-α).
LINE 43 aggravation of insulin resistance - not a correct term, worsening, increase, progression instead aggravation should be used.
LINE 45 Untangle this sentance/divide in two: Free fatty acids released from adipocytes flow through blood to the liver and contribute to hepatic steatosis by inducing de novo lipogenesis resulted from decreased mitochondrial β-oxidation of fat and increasing the synthesis of triglycerides.
LINE 53 - define which AO patterns in the introduction
LINE 55 However, the relationship between AO patterns and the incidence of NAFLD has not been verified. - This is simply not true, I believe you overgeneralized in this sentance. Specifiy what exactly is "not verified", or not understood enough. There is plenty evidence investigatin correlation of NAFLD and different AO "patterns" in differently designed studies. Reformulate the sentance - simply as you explained at the end of the discussion (LINE 331).
Line 122-124 - separate this formula as a figure or as an equation (available instructions in mdpi instructions for authors).
LINE 148 - would suggest a better term for improved from AO - to just improved AO or another term.
LINE 157 - as previously mentioned, use mdpi equation implementation instructions to separate the equations, throught the manuscript for a better organization of the paper.
Tables 1 and 2 should be organized better, the right column is really confusing, perhaps separate the rows with borders, as you did in Table 3.
Conclusion - although these results are valuable, and study is overall a contribution to the field, conclusion does not say much. Include precise recommendations, would pharmacotherapy be worth examining, what could be further investigated, perhaps physical activity, pharmacotherapy, etc. among these groups and how it would affect the results and contribute to ameliorating NAFLD.
Well written, minor editing required.
Author Response
Response to Reviewer #3
The manuscript is well written. All sections appropriately provide detailed information, except conclusion (see further comments). Methods and results are well described, tehnical editing is required (explained in further comments). Researchers provided an interesting insight into correlation of NAFLD and abdominal obesity.
In abstract (an throught the study) consider a better term for a "improved from AO", perhaps just "improved AO"
Response: Thank you for your valuable comment. We have revised "improved from AO" to "improved AO" throughout the manuscript.
LINE 42 Decode this as interleukin 6 (IL-6) and tumor necrosis factor α (TNF-α).
Response: Thank you for your valuable comment. We have revised this as recommended.
|
Excessive visceral adipose tissue (VAT) promotes secretion of pro-inflammatory cytokines such as interleukin 6 (IL-6) and tumor necrosis factor α (TNF-α), which leads to the development and worsening of insulin resistance [4]. |
LINE 43 aggravation of insulin resistance - not a correct term, worsening, increase, progression instead aggravation should be used.
Response: Thank you for your valuable comment. We have modified this as recommended.
|
Excessive visceral adipose tissue (VAT) promotes secretion of pro-inflammatory cytokines such as interleukin 6 (IL-6) and tumor necrosis factor α (TNF-α), which leads to the development and worsening of insulin resistance [4]. |
LINE 45 Untangle this sentance/divide in two: Free fatty acids released from adipocytes flow through blood to the liver and contribute to hepatic steatosis by inducing de novo lipogenesis resulted from decreased mitochondrial β-oxidation of fat and increasing the synthesis of triglycerides.
Response: Thank you for your valuable comment. We have revised this sentence as recommended.
|
Free fatty acids released from adipocytes flow through the blood to the liver and contribute to hepatic steatosis [6]. This contribution occurs by inducing de novo lipogenesis, resulting from decreased mitochondrial β-oxidation of fat and increasing the synthesis of triglycerides [6]. |
LINE 53 - define which AO patterns in the introduction
Response: We apologize for the confusing expression. We have modified as follows:
|
Since AO status can change over time, it is more important to consider the long-term trends in AO status rather than spot-checked AO status to determine the risk for metabolic diseases. |
LINE 55 However, the relationship between AO patterns and the incidence of NAFLD has not been verified. - This is simply not true, I believe you overgeneralized in this sentance. Specifiy what exactly is "not verified", or not understood enough. There is plenty evidence investigatin correlation of NAFLD and different AO "patterns" in differently designed studies. Reformulate the sentance - simply as you explained at the end of the discussion (LINE 331).
Response: I apologize for the inaccurate statement. I appreciate your correction. I have rephrased the sentence for clarity as follows:
|
While there is significant evidence exploring the correlation between different patterns of AO and the incidence of NAFLD in various studies, implications of this relationship may require further investigation and a deeper understanding. |
Line 122-124 - separate this formula as a figure or as an equation (available instructions in mdpi instructions for authors).
Response: Thank you for your comment. I have separated this formula as an equation.
LINE 148 - would suggest a better term for improved from AO - to just improved AO or another term.
Response: Thank you for your valuable comment. We have changed this term as recommended.
LINE 157 - as previously mentioned, use mdpi equation implementation instructions to separate the equations, throught the manuscript for a better organization of the paper.
Response: Thank you for your comment. I have separated this formula as an equation.
DM (Yes: 2, No: 0)
Tables 1 and 2 should be organized better, the right column is really confusing, perhaps separate the rows with borders, as you did in Table 3.
Response: We apologize for the confusing expression. We have separated the rows with borders in Table 2 and Table 3.
Conclusion - although these results are valuable, and study is overall a contribution to the field, conclusion does not say much. Include precise recommendations, would pharmacotherapy be worth examining, what could be further investigated, perhaps physical activity, pharmacotherapy, etc. among these groups and how it would affect the results and contribute to ameliorating NAFLD.
Response: Thank you for your valuable comment. We revised the conclusion section as follows;
|
Line 380 In total population, persistent AO and progression to AO are associated with higher risk for NAFLD. Persistent AO was a significant risk factor for developing NAFLD only in women, suggesting that both maintaining lean WC or improvement from AO would be effective strategies for preventing NAFLD in women while maintaining lean WC is more crucial in men. A health strategy that focuses on maintaining a healthy WC throughout life through physical activity and a healthy diet is likely to be more effective in preventing NAFLD than solely relying on reducing WC at a later stage. Also, considering gender-specific risk profiles can ultimately contribute to the development of more effective health policies and strategies for addressing NAFLD and related health concerns. Additional research is warranted to comprehensively assess the severity of NAFLD, in order to obtain a more precise understanding of the relationship between AO and the risk of NAFLD. |

Round 2
Reviewer 2 Report
I agree with the publication.